# Online Compatible Reward Identification from Preference Feedback

**Simone Drago** [1]   **Marco Mussi** [1]   **Alberto Maria Metelli** [1]

## Abstract

In reinforcement learning, human preference feedback is emerging as a viable alternative to expert-designed reward functions, which can be difficult to engineer in real-world problems. However, despite the growing importance of preference feedback, how to effectively *elicit* preferences remains a fundamental open problem. This work focuses on the *compatible reward identification* task. The aim is to derive, starting from preference feedback, a reward function compatible with the observed preferences and accurate across the *entire* state-action space, ensuring higher transferability, safety, and interpretability. Indeed, the most common *reinforcement learning from human feedback* objective is to learn the *optimal policy*, requiring accuracy only in the portion of the state-action space that the agent visits. However, this goal cannot provide the same guarantees as compatible reward identification. First, we discuss commonalities and differences between the two goals. Then, we consider deterministic preferences, deriving the minimum number of interactions needed to identify the set of compatible rewards, and showing that using fewer queries may lead to arbitrarily large suboptimality. Finally, we focus on stochastic preferences generated via the Bradley-Terry (BT) model. We introduce the concepts of *query basis* and its *index*, relating them to the problem complexity. Upon this, we discuss the connection between the index of a basis and the BT model, as well as the limitations that the model induces in this setting. Additionally, we devise an algorithm to identify a nearly-optimal query basis with polynomial human query complexity.

## 1. Introduction

Throughout the years, machine learning approaches have proven to be incredibly successful in addressing a large variety of real-world problems. This success can be attributed to the effective use of powerful learning paradigms, capable of capturing the distinctive features that characterize the decision-making processes involved. Among these, *reinforcement learning* (RL, Sutton & Barto, 2018) is considered the paradigm most closely aligned with the learning process of human beings. RL has shown remarkable successes, sometimes even surpassing human performance, in domains such as autonomous driving (Kiran et al., 2021), finance (Zhang et al., 2020), and industrial plant control (Kober et al., 2013).

The defining characteristic of RL is the use of a *reward* function, which is considered "the most succinct, robust, and transferable definition of a task" (Abbeel & Ng, 2004). Beyond defining the objective, which can prove to be difficult and limited by itself (Sutton, 2004; Silver et al., 2021; Glukhov, 2022), the reward function should also encode the *intent* (Ng & Russell, 2000) of the learner. The process of devising a reward function is called *reward engineering* (Dewey, 2014). It is well known that it is a highly challenging task (Ng & Russell, 2000), often prone to mistakes that can compromise the performance of the learned behavior (Ng et al., 1999). Ideally, the system designer would like to reconstruct the reward function that guides the human decision-making process in the specific task at hand, assuming that such a function exists. However, humans do not *explicitly communicate* any underlying reward function that guided their decisions. This observation suggests that the use of a hand-crafted *reward feedback* should be reconsidered when attempting to align RL more closely with the human decision-making process, ultimately scaling RL to real-world problems in which such an alignment is a critical requirement.

Recently, forms of *feedback* other than the reward have been considered for training learning agents. Although multiple types of feedback exist (Jeon et al., 2020), including corrections, improvements, indications, and proxy rewards, *human-in-the-loop RL* (Retzlaff et al., 2024) has emerged as a key paradigm for training RL agents using human-generated *preference feedback*. Intuitively, asking a human

---

[1]Politecnico di Milano, Milan, Italy.
Correspondence to: Simone Drago <simone.drago@polimi.it>.

*Proceedings of the 43rd International Conference on Machine Learning*, Seoul, South Korea. PMLR 306, 2026. Copyright 2026 by the author(s).

to choose between two (or more) options is more realistic than asking them to assign a precise *numerical reward* to each option. While *preference-based RL* (Fürnkranz et al., 2012; Wirth et al., 2017) has a long-standing history in the literature, it has recently gained renewed attention due to the rise of large language models (Zhao et al., 2023).

The first step towards defining a preference-based approach is to model how humans generate preferences, i.e., the *human rationality model*. The most common modeling choice follows the *utility representation theorem* (von Neumann & Morgenstern, 1947), which postulates that any rational agent generates their preferences in terms of an underlying utility function defined over the available options. In the context of RL, as customary in the literature, this translates into assuming the existence of a utility function $u$ that assigns a scalar value to each trajectory, and considering preferences defined over pairs of trajectories (see, e.g., Christiano et al., 2017). Additionally, it is commonly assumed that the utility function corresponds to the sum of rewards along the trajectory, whenever a Markovian reward is available or definable. In the literature, the preference generation process derived from the utility representation theorem is defined via a *probabilistic model*. Given two trajectories $\tau, \tau' \in \mathcal{T}$, the probability that $\tau$ is preferred over $\tau'$, denoted as $\tau > \tau'$, is defined in terms of their respective utilities as:

$$\mathbb{P}(\tau > \tau') = g(u(\tau) - u(\tau')), \qquad (1)$$

where $g : \mathbb{R} \to [0, 1]$ is a *monotonically non-decreasing* function such that $g(x) + g(-x) = 1$, for every $x \in \mathbb{R}$. When presented with a pair of trajectories $(\tau, \tau')$, that from here onward we will refer to as *query*, the human labeler provides a binary feedback that can be interpreted as a sample from a Bernoulli random variable with parameter $\mathbb{P}(\tau > \tau')$, with outcome 1 representing the preference of $\tau$ over $\tau'$, and outcome 0 representing the opposite preference. Intuitively, the greater the difference between $\tau$ and $\tau'$ in terms of utilities, the higher the probability of observing $\tau > \tau'$. We consider two interesting choices for function $g$: (*i*) $g(\cdot) := \sigma(\cdot)$, i.e., the sigmoid function $\sigma(x) = 1/(1 + e^{-x})$ for $x \in \mathbb{R}$, which leads to the renowned *Bradley-Terry* (BT, Bradley & Terry, 1952) model customarily used to characterize stochastic preferences; and (*ii*) $g(\cdot) := \mathbb{H}(\cdot)$, i.e., the *Heaviside step function* (Abramowitz & Stegun, 1972), taking values:

$$\mathbb{H}(x) := \begin{cases} 1 & x > 0, \\ \frac{1}{2} & x = 0, \\ 0 & x < 0, \end{cases}$$

satisfying Equation (1), and leading to deterministic preferences.

In this work, we focus on the *compatible reward identification* problem. The objective of the learner is to minimize the *query sample complexity*, i.e., the number of interactions with a human labeler necessary to recover a reward function

that is *compatible*, meaning that its trajectory return can be used to model the human labeler preferences for every pair of trajectories.[1] Achieving this objective requires recovering a reward function that is accurate on the *entire* state-action space. As thoroughly discussed in the *inverse reinforcement learning* (IRL, Abbeel & Ng, 2004) literature, such a high degree of accuracy provides generalizability and transferability guarantees of the learned behavior (Arora & Doshi, 2021), which are particularly relevant, e.g., in sim-to-real scenarios (Lazzati & Metelli, 2026). Additionally, we consider both *offline* and *online* scenarios, demonstrating that feedback can be leveraged to reduce sample complexity.

**Original Contributions.** In this paper, we take a step toward a theoretical understanding of *exploration* in preference-based RL. To this end, we consider *tabular* environments, assuming that the utility function $u$ is defined as the trajectory return of an unknown reward function $r^*$, allowing us to focus entirely on the problem of reward identification from preference feedback. In this setting, we define an algorithm that, given a likelihood function $g$, selects a sequence of trajectory pairs $(\tau, \tau')$ to present to the human with the goal of recovering a reward function compatible with $r^*$, minimizing the query complexity. Specifically, our contributions are as follows:

- Regarding the case of *deterministic preferences*, i.e., when $g(\cdot) = \mathbb{H}(\cdot)$, we provide lower bounds on the number of samples required to recover a compatible reward function given a *sufficient* query set in both the offline and online scenarios. Furthermore, we prove an *inapproximability* result, showing that using a number of queries lower than the lower bound can lead to arbitrarily large suboptimality in the recovered reward function $\widehat{r}$.

- Regarding the case of *stochastic preferences*, where we consider the Bradley-Terry model, i.e., when $g(\cdot) = \sigma(\cdot)$, we characterize the quality of queries in terms of two competing factors: (*i*) the information-theoretic value for recovering the unknown reward $r^*$, and (*ii*) the sample complexity required to reliably estimate $r^*$. Upon this, we propose an $(\varepsilon, \delta)$-PAC algorithm that identifies a *nearly-optimal set of queries* and recovers an $\varepsilon$-compatible reward $\widehat{r}$, with probability at least $1 - \delta$.

We provide omitted proofs in Appendix A. In Appendix B, we provide a comparison of the pros and cons of our objective as opposed to the more common *policy optimization* objective. We present a compatible reward identification algorithm, together with numerical simulations, in Appendix C.

---

[1] We use the keyword *compatible* to represent the existence of multiple reward functions that are compliant with the feedback we receive. We formally define *compatible reward* in Section 2.

## 2. Setting

In this section, we introduce the setting, the interaction protocol, and the learning problem.

**Notation.** Let $a, b \in \mathbb{N}$ such that $a < b$, we define $[\![a, b]\!] := \{a, a+1, \ldots, b\}$ and $[\![a]\!] := [\![1, a]\!]$. Let $\mathcal{X}$ be a finite set, we denote with $\Delta(\mathcal{X})$ the probability simplex over the set, and with $|\mathcal{X}|$ its cardinality.

**Markov Decision Processes.** A finite-horizon Markov Decision Process (MDP, Puterman, 2014) is defined as a tuple $\langle \mathcal{S}, \mathcal{A}, p, r, \mu, H \rangle$, where $\mathcal{S}$ and $\mathcal{A}$ are the finite (i.e., $|\mathcal{S}| =: S < +\infty$ and $|\mathcal{A}| =: A < +\infty$) state and action spaces, $p : \mathcal{S} \times \mathcal{A} \times [\![H]\!] \to \Delta(\mathcal{S})$ is the (stage-dependent) transition probability, $r : \mathcal{S} \times \mathcal{A} \times [\![H]\!] \to [0, R_{\max}]$ is the (stage-dependent) reward function, $\mu \in \Delta(\mathcal{S})$ is the initial-state distribution, and $H \in \mathbb{N}$ is the length of an episode. A *trajectory* $\tau := (s_h, a_h)_{h \in [\![H]\!]}$ is a sequence of $H$ state-action pairs. We denote as $\mathcal{T} := (\mathcal{S} \times \mathcal{A})^H$ the set of possible trajectories with cardinality $|\mathcal{T}| = (SA)^H$. A *utility function* $u : \mathcal{T} \to \mathbb{R}$ maps every trajectory $\tau \in \mathcal{T}$ to a real value $u(\tau) \in \mathbb{R}$. The reward function $r$ induces the *trajectory return* $u_r$, i.e., the utility defined as the sum of the rewards of state-action pairs in the trajectory:

$$u_r(\tau) := \sum_{h \in [\![H]\!]} r_h(s_h, a_h), \quad \forall \tau = (s_h, a_h)_{h \in [\![H]\!]} \in \mathcal{T}. \quad (2)$$

**Preference Generation.** The *interaction* between the agent and the human labeler proceeds as follows. At each round of interaction $t \in \mathbb{N}$, the human labeler is presented with a pair of trajectories $(\tau_t, \tau'_t) \in \mathcal{T} \times \mathcal{T}$, is asked to express a preference, and provides a feedback $y_t \in \{0, 1\}$ (1 if $\tau_t > \tau'_t$ and 0 otherwise), with $y_t \sim \text{Ber}(\mathbb{P}(\tau_t > \tau'_t))$, where $\text{Ber}(p)$ represents the Bernoulli distribution with parameter $p \in [0, 1]$. As per Equation (1), we define the probability of observing a preference, i.e., $\mathbb{P}(\tau_t > \tau'_t)$, in terms of a function $g$ and a utility $u$. We now formally define the pair $(\mathbb{P}, g)$.

**Definition 2.1** (Preference Model). *A preference model is a pair* $(g, u)$*, where* $g \in \mathfrak{G} := \{\mathfrak{g} : \mathbb{R} \to [0, 1] : \mathfrak{g}(x) + \mathfrak{g}(-x) = 1 \ \forall x \in \mathbb{R} \wedge \mathfrak{g} \ \text{non-decreasing}\}$ *is a* likelihood function *and* $u \in \mathfrak{U} := \{\mathfrak{u} : \mathcal{T} \to \mathbb{R}\}$ *is a* utility function. *Given a reward function* $r \in \mathfrak{R} := \{\mathfrak{r} : \mathcal{S} \times \mathcal{A} \times [\![H]\!] \to \mathbb{R}\}$*, we denote* $(g, r) := (g, u_r)$*, where* $u_r$ *is the trajectory return induced by* $r$.

Definition 2.1 induces a probability distribution over trajectory pairs $\mathbb{P} : \mathcal{T} \times \mathcal{T} \to \Delta(\{0, 1\})$ defined according to Equation (1). We assume the likelihood function is fixed and given prior to the learning process, whereas the utility (or reward) function is the output of the learning process, to be formally defined later.

Following the utility representation theorem (von Neumann & Morgenstern, 1947), we assume the existence of an underlying utility function $u^* \in \mathfrak{U}$ guiding human preferences. However, as we will discuss in Section 3, the identification of a utility function is statistically intractable; thus, we require the additional assumption of the existence of an underlying reward function $r^* \in \mathfrak{R}$. Intuitively, when considering the rationality model of Equation (1), there may exist several rewards $r$ that induce the same probability distribution $\mathbb{P}$. Thus, we define a *compatible reward set* as the set of all reward functions that are equivalent w.r.t. the goal of recovering the human preference model, i.e., they induce the same probability distribution over trajectory pairs.

**Definition 2.2** (Compatible Reward Set). *Given a likelihood function* $g \in \mathfrak{G}$ *and a probability distribution* $\mathbb{P} : \mathcal{T} \times \mathcal{T} \to \Delta(\{0, 1\})$*, we define as* compatible reward set $\mathcal{R}(\mathbb{P}, g)$ *the set of all reward functions* $r \in \mathfrak{R}$ *such that the preference model* $(g, r)$ *induces* $\mathbb{P}$:

$$\mathcal{R}(\mathbb{P}, g) := \{r \in \mathfrak{R} : \mathbb{P}(\tau > \tau') = g(u_r(\tau) - u_r(\tau')),$$
$$\forall (\tau, \tau') \in \mathcal{T} \times \mathcal{T}\}.$$

Clearly, we have that $r^* \in \mathcal{R}(\mathbb{P}, g)$; however, depending on the likelihood function, many other rewards are present in the set. For instance, with the BT model, we have *translation invariance* (i.e., $r^* + \beta \in \mathcal{R}(\mathbb{P}, g)$ for any $\beta \in \mathbb{R}$). Regarding the deterministic model, instead, we have more degrees of freedom, e.g., *positive scale-translation invariance* (i.e., $\alpha r^* + \beta \in \mathcal{R}(\mathbb{P}, g)$, for every $\alpha > 0$ and $\beta \in \mathbb{R}$). Let us now formalize the learning objective.

**Definition 2.3** (Instance of CRI-Tf Problem). *A compatible reward identification from trajectory-preference feedback (CRI-Tf) problem instance is a pair* $(\mathbb{P}, g)$*, where* $\mathbb{P} : \mathcal{T} \times \mathcal{T} \to \Delta(\{0, 1\})$ *is a probability distribution over trajectory pairs and* $g \in \mathfrak{G}$ *is a likelihood function.*

Given an instance $(\mathbb{P}, g)$ of the CRI-Tf problem, the goal of the agent is to learn a reward function that lies in (an approximation of) $\mathcal{R}(\mathbb{P}, g)$.

## 3. Queries and Learning Problem

In this section, we formalize the notion of *query* and introduce the learning problem.

**Queries.** Before defining the notion of query, we introduce the auxiliary concept of *binary trajectory representation*. Let $\tau = (s_h, a_h)_{h \in [\![H]\!]} \in \mathcal{T}$ be a trajectory, we can represent it via a binary vector $\mathbf{b}(\tau) \in \{0, 1\}^{SAH}$ whose entries are 1 only in the state-action-stage triplets that appear in $\tau$. Specifically, denoting as $b(\tau)_{s, a, h}$ the element of $\mathbf{b}(\tau)$ that corresponds to state $s$, action $a$, and stage $h$, we have that:

$$b(\tau)_{s, a, h} = \begin{cases} 1 & \text{if } (s_h, a_h) = (s, a), \\ 0 & \text{otherwise,} \end{cases}$$

for every $(s, a, h) \in \mathcal{S} \times \mathcal{A} \times [\![H]\!]$ and $(s_h, a_h) \in \tau$. We observe that $\sum_{s,a \in \mathcal{S} \times \mathcal{A}} b(\tau)_{s,a,h} = 1$ for every $h \in [\![H]\!]$ and $\tau \in \mathcal{T}$. Additionally, we denote the set of binary vectors that represent all trajectories as $\mathcal{B} = \{\mathbf{b}(\tau) : \tau \in \mathcal{T}\}$.

We define a pair of trajectories $(\tau, \tau') \in \mathcal{T} \times \mathcal{T}$ as a *query* to present to the human, and we represent it via the ternary vector $\mathbf{q}(\tau, \tau') := \mathbf{b}(\tau) - \mathbf{b}(\tau') \in \{-1, 0, 1\}^{SAH}$, which we call a *query vector*. Given a query set $\mathcal{Q} = \{(\tau_1, \tau_1'), \ldots, (\tau_Q, \tau_Q')\} \subseteq \mathcal{T} \times \mathcal{T}$, with $|\mathcal{Q}| =: Q$, we represent it via the *query matrix*, i.e., the ternary matrix $\mathbf{Q} = (\mathbf{q}(\tau_i, \tau_i')^\top)_{i \in [\![Q]\!]} \in \{-1, 0, 1\}^{Q \times SAH}$ obtained by stacking the query vectors $\mathbf{q}(\tau_i, \tau_i')$ of queries in $\mathcal{Q}$.

Note that two distinct queries $(\tau_1, \tau_1') \neq (\tau_2, \tau_2')$ may lead to the same query vector, i.e., $\mathbf{q}(\tau_1, \tau_1') = \mathbf{q}(\tau_2, \tau_2')$, in which case we say that they are *equivalent*, and we denote this as $(\tau_1, \tau_1') \approx (\tau_2, \tau_2')$. Intuitively, equivalence between queries means that they *leak* the same information about the underlying reward function.

Given a likelihood function $g \in \mathfrak{G}$ and a query set $\mathcal{Q} \subseteq \mathcal{T} \times \mathcal{T}$, we introduce the *$\mathcal{Q}$-compatible reward set* as the set of all reward functions $r$ such that the preference model $(g, r)$ induces the true probability distribution $\mathbb{P}$ restricted to the queries in $\mathcal{Q}$:

$$\mathcal{R}(\mathbb{P}, g; \mathcal{Q}) := \{r \in \mathfrak{R} : \mathbb{P}(\tau \succ \tau') = g(u_r(\tau) - u_r(\tau')),$$
$$\forall (\tau, \tau') \in \mathcal{Q}\}.$$

Intuitively, given two query sets $\mathcal{Q}, \mathcal{Q}' \subseteq \mathcal{T} \times \mathcal{T}$ such that $\mathcal{Q} \subseteq \mathcal{Q}'$, it holds that $\mathcal{R}(\mathbb{P}, g; \mathcal{Q}) \supseteq \mathcal{R}(\mathbb{P}, g; \mathcal{Q}')$. Moreover, we notice that $\mathcal{R}(\mathbb{P}, g; \{\}) = \mathfrak{R}$ is the set of all possible reward functions and $\mathcal{R}(\mathbb{P}, g; \mathcal{T} \times \mathcal{T}) = \mathcal{R}(\mathbb{P}, g)$ is the set of reward functions compatible with $r^*$.

**Learning Problem.** Given an instance $(\mathbb{P}, g)$ of the CRI-Tf problem, the goal of the agent is to gather the *minimum amount of information*, in the form of human preferences over trajectory pairs, that allows it to identify a reward $\widehat{r} \in \mathfrak{R}$ that lies in the set of compatible rewards, i.e., $\widehat{r} \in \mathcal{R}(\mathbb{P}, g)$. To this end, we introduce the notion of *sufficient query set*, i.e., a query set that allows the identification of $\mathcal{R}(\mathbb{P}, g)$.

**Definition 3.1** (Sufficient, Minimal, and Minimum Query Sets). *Let $(\mathbb{P}, g)$ be a preference model and $\mathcal{Q} \subseteq \mathcal{T} \times \mathcal{T}$ be a query set. $\mathcal{Q}$ is* sufficient *if $\mathcal{R}(\mathbb{P}, g; \mathcal{Q}) = \mathcal{R}(\mathbb{P}, g)$. Moreover, a sufficient query set $\mathcal{Q}$ is: $(i)$* minimal *if there exists no strict subset $\mathcal{Q}' \subset \mathcal{Q}$ that is sufficient, and $(ii)$* minimum *if there exists no sufficient query set with strictly smaller cardinality than $|\mathcal{Q}|$.*

Clearly, a minimum query set is also minimal.[2] We refer to minimum query sets as *bases*.

---

[2]Intuitively, the sufficiency of a query set is a property that depends on both the likelihood function $g$ and the probability distribution $\mathbb{P}$ over trajectory pairs.

In the following sections, we consider both *offline* algorithms, which choose the query set prior to interacting with the human labeler, and *online* algorithms, which select the next query based on the feedback collected up to that point.

Any CRI-Tf algorithm outputs a reward function $\widehat{r} \in \mathcal{R}(\mathbb{P}, g; \widehat{\mathcal{Q}}) \subseteq \mathfrak{R}$. For ease of notation, we refer to $\mathcal{R}(\mathbb{P}, g)$ as $\mathcal{R}$ and to $\mathcal{R}(\mathbb{P}, g; \widehat{\mathcal{Q}})$, where $\widehat{\mathcal{Q}}$ is the query set chosen by a CRI-Tf algorithm, as $\widehat{\mathcal{R}}$. Following the standard practice of IRL (Metelli et al., 2021; Lazzati et al., 2024), we evaluate the distance between the two sets $\mathcal{R}, \widehat{\mathcal{R}} \subseteq \mathfrak{R}$ in terms of *Hausdorff distance* (Rockafellar & Wets, 2009), defined as:

$$\mathcal{H}(\mathcal{R}, \widehat{\mathcal{R}}) := \max\Big\{\sup_{r \in \mathcal{R}} \inf_{\widehat{r} \in \widehat{\mathcal{R}}} \|r - \widehat{r}\|_\infty, \sup_{\widehat{r} \in \widehat{\mathcal{R}}} \inf_{r \in \mathcal{R}} \|r - \widehat{r}\|_\infty\Big\}.$$
(3)

Intuitively, it represents the *minimum worst-case* error we can make in recovering a compatible reward.

Finally, to evaluate the performance of a CRI-Tf algorithm, we use the standard *probably approximately correct* (PAC) definition.

**Definition 3.2** (PAC CRI-Tf Algorithm). *Let $\varepsilon \geq 0$ and $\delta \in [0, 1)$, a CRI-Tf algorithm is $(\varepsilon, \delta)$-PAC if, for every CRI-Tf problem instance $(\mathbb{P}, g)$, it generates a query set $\widehat{\mathcal{Q}}$ such that $\Pr(\mathcal{H}(\mathcal{R}, \widehat{\mathcal{R}}) > \varepsilon) \leq \delta$. We denote its cardinality $|\widehat{\mathcal{Q}}|$ as* query complexity.

Thus, an $(\varepsilon, \delta)$-PAC algorithm generates a query set $\widehat{\mathcal{Q}}$, inducing a $\widehat{\mathcal{Q}}$-compatible set of rewards that allows to identify the true compatible set of rewards $\mathcal{R}(\mathbb{P}, g)$ up to an error $\varepsilon$ expressed in terms of Hausdorff distance, with probability at least $1 - \delta$. Our goal is to design $(\varepsilon, \delta)$-PAC algorithms that minimize the query complexity.

**On the Need for Markovian Rewards.** We now briefly motivate *why* from Section 2 onward we consider Markovian rewards and their trajectory return, i.e., Equation (2), rather than directly working with general (possibly non-Markovian) utility functions. For general utility functions, it is impossible to devise a CRI-Tf algorithm that complies with Definition 3.2 and has a cardinality that is *polynomial* in the problem parameters. Trivially, in such a case, it is necessary to observe each trajectory at least once in a query, since learning utilities that induce an *incorrect ordering* of trajectories can lead to arbitrarily large Hausdorff distance, as we will formally demonstrate later on in Theorem 4.3. Indeed, since $|\mathcal{T}| = (SA)^H$, this leads to an exponential dependency on $H$, which is undesirable and makes the problem intractable. Thus, the need to work with Markovian rewards. In principle, even if we know that preferences are guided by a non-Markovian utility, it is possible to solve a surrogate, approximate Markovian problem at the cost of a slight performance suboptimality (see, e.g., Drago et al., 2025, Theorem 6.1).

# 4. Deterministic Preferences

In this section, we focus on the case of deterministic preferences, i.e., when $g = \mathbb{H}$. By plugging this likelihood function into Equation (1), we can rewrite it as $\mathbb{P}(\tau > \tau') = \mathbb{H}(u(\tau) - u(\tau'))$. Section 4.1 discusses approximability issues in recovering a compatible reward set with deterministic preferences.

## 4.1. Lower Bounds and Approximability

In the case of the deterministic likelihood model $g = \mathbb{H}$, the goal of the agent is to choose a query set $\mathcal{Q}$, either online or offline, to recover (an approximation of) the compatible reward set $\mathcal{R}(\mathbb{P}, \mathbb{H})$.

We now report query complexity *lower bounds* (LBs) for both the offline and online cases, under deterministic preference generation. In the offline case, the agent chooses the *entire* query set prior to interacting with the human labeler.

**Theorem 4.1** (Query Complexity LB - Offline). *Consider a $(0,0)$-PAC offline algorithm. Then, there exists an MDP and a CRI-Tf instance $(\mathbb{P}, \mathbb{H})$ such that there exists no* sufficient *query set with query complexity less than $HSA(SA - 1)/2$.*

This result follows from observing that, to recover a compatible reward function, it is at least required to identify a compatible reward that is compatible with each stage independently of the others. This corresponds to finding the total order among state-action pairs induced by the true reward function at every stage. Indeed, in the offline case, it is intuitive to observe that, if we do not label every unordered query, there exists an MDP in which we may recover the wrong total order in the worst case. One could be tempted to overlook this problem and accept an approximation of the reward in the case of an incorrect ordering of two state-action pairs. However, in Theorem 4.3, we will demonstrate that this can induce an arbitrarily large Hausdorff distance.

Moving to the online case, where the agent chooses a query based on previously collected information, we aim to exploit past feedback to reduce the number of queries. The following result suggests this is the case.

**Theorem 4.2** (Query Complexity LB - Online). *Consider a $(0,0)$-PAC online algorithm. Then, there exists an MDP and a CRI-Tf instance $(\mathbb{P}, \mathbb{H})$ such that there exists no* sufficient *query set with query complexity less than $\Omega(HSA \log_2(SA))$.*

This result is proved by reducing to the *sorting* problem. Indeed, it is not necessary to verify all unordered queries, as we can exploit the transitivity of the deterministic likelihood. This corresponds to sorting state-action pairs in every stage to recover an instantaneous reward compatible for that stage. It is well-known that sorting $SA$ elements takes at least $\Omega(SA \log_2(SA))$ queries, i.e., comparisons (Knuth, 1998).

We obtain the result by repeating this for every stage.

Theorems 4.1 and 4.2 follow the intuition that the ability to select what queries to show the labeler based on the knowledge gathered up to that point, i.e., online, may reduce the overall query complexity.

We now present an *inapproximability* result showing that for every non-sufficient query set, there exists an MDP where it leads to the maximum possible Hausdorff distance.

**Theorem 4.3** (Inapproximability). *Let $\mathcal{S}$ be a state space, $\mathcal{A}$ an action space, and $H \in \mathbb{N}$ be a horizon. Consider the likelihood function $g = \mathbb{H}$. For every query set $\mathcal{Q}$ that is minimal for some reward function $r \in \mathfrak{R}$ (i.e., every state-action-stage triple appears in one of its trajectories) and query $(\tau, \tau') \in \mathcal{Q}$, there exists an MDP $\langle \mathcal{S}, \mathcal{A}, p, r, \mu, H \rangle$ such that $\mathcal{H}(\mathcal{R}(\mathbb{P}, \mathbb{H}; \mathcal{Q} \backslash \{(\tau, \tau')\}), \mathcal{R}(\mathbb{P}, \mathbb{H})) = R_{\max}$, where $\mathbb{P}$ is the probability distribution induced by $r$.*

The proof follows from observing that, whenever we consider $\mathcal{Q} \backslash \{(\tau, \tau')\}$, there exists an MDP in which we are unable to infer whether $\tau > \tau'$ or $\tau' > \tau$. This, in turn, implies that the compatible reward set can be partitioned into two disjoint sets, each inducing one of the two orderings. By putting all the reward mass on one of the two trajectories, we demonstrate that in the worst case, it holds that:

$$\mathcal{H}(\mathcal{R}(\mathbb{P}, \mathbb{H}; \mathcal{Q}), \mathcal{R}(\mathbb{P}, \mathbb{H}; \mathcal{Q} \backslash \{(\tau, \tau')\})) = R_{\max}.$$

This result shows that attempting to identify a compatible reward function using a non-sufficient query set leads to a worst-case maximum Hausdorff distance. Thus, Theorem 4.3 forces any learning algorithm to devise sufficient query sets only. As a consequence, an $(\varepsilon, \delta)$-PAC algorithm, in the case of the deterministic likelihood function, can only attain either $\varepsilon = 0$ if it chooses a sufficient query set, or $\varepsilon = R_{\max}$ for at least some problem instances when it chooses a non-sufficient query set. Since no stochasticity is involved in the preference generation process with $g = \mathbb{H}$, we can safely choose $\delta = 0$.

# 5. Stochastic Preferences

In this section, we focus on the case of stochastic preferences, i.e., when $g = \sigma$. By plugging this likelihood model into Equation (1) under the Markovian rewards assumption, we can rewrite it as $\mathbb{P}(\tau > \tau') = \sigma(u_r(\tau) - u_r(\tau'))$. In Section 5.1, we briefly consider the simplistic case in which we have direct access to $\mathbb{P}(\tau > \tau')$. This allows introducing some core concepts employed in Section 5.2 to address the case in which we observe samples from a Bernoulli with parameter $\mathbb{P}(\tau > \tau')$. In the remainder of this section, except when stating complexity results, we omit stage $h$ from the notation, as we consider trajectory pairs that differ for a single state-action pair.

## 5.1. Probability Feedback

Let us first consider the case in which, upon presenting a query $(\tau, \tau')$ to the human labeler, we are given the probability of observing the preference, i.e., $\mathbb{P}(\tau > \tau')$.[3] Similarly to the deterministic case, there is no stochasticity in the feedback we obtain, and thus we can aim to devise $(0,0)$-PAC algorithms.

**Theorem 5.1** (Query Complexity LB – Probability Feedback). *Consider a $(0,0)$-PAC offline algorithm. Then, there exists an MDP and a CRI-Tf instance $(\mathbb{P}, \sigma)$ such that the query complexity is at least $H(SA - 1)$.*

*Proof.* The proof follows by observing that, under the BT model, we can compute the difference in terms of reward between any two state-action pairs for any stage $h \in [\![H]\!]$ via the inverse sigmoid:

$$
\begin{aligned}
\sigma^{-1}&(\mathbb{P}(\tau \xleftarrow{h} (s,a) > \tau \xleftarrow{h} (s',a'))) \\
&= \sigma^{-1}(\sigma(u_r(\tau \xleftarrow{h} (s,a)) - u_r(\tau \xleftarrow{h} (s',a')))) \\
&= u_r(\tau \xleftarrow{h} (s,a)) - u_r(\tau \xleftarrow{h} (s',a')) \\
&= r_h(s,a) - r_h(s',a'),
\end{aligned}
$$

where $\tau \xleftarrow{h} (s,a)$ represents modifying trajectory $\tau$ by changing the state-action pair at stage $h$ with $(s,a)$. Then, in order to identify $\mathcal{R}(\mathbb{P}, \sigma)$ we require $(SA - 1)$ independent equations for each $h \in [\![H]\!]$. Under the BT model, we have $H$ degrees of freedom corresponding to the translation invariance for every stage. $\square$

**Algorithm.** Let us denote as $\mathfrak{B}$ the set of bases, as characterized in Definition 3.1. Denote with $\mathfrak{C} \subseteq \mathfrak{B}$ the set of bases whose matrix representation, for any given stage, is an upper bidiagonal matrix containing 1 on the main diagonal, $-1$ directly above it, and 0 elsewhere, up to a permutation of queries and state-action pairs. We will refer to query sets in $\mathfrak{C}$ as *query chains*.

**Example 1** (Query Chain). *Consider an MDP with $\mathcal{S} = \{s_1, s_2\}$, $\mathcal{A} = \{a_1, a_2\}$, and $H = 1$. Then, $\mathcal{T} = \{\tau_1 := (s_1, a_1), \tau_2 := (s_1, a_2), \tau_3 := (s_2, a_1), \tau_4 := (s_2, a_2)\}$. Query set $\mathcal{Q}_1 = \{(\tau_1, \tau_2), (\tau_1, \tau_3), (\tau_3, \tau_4)\}$ is clearly a basis, but not a query chain. Instead, query set $\mathcal{Q}_2 = \{(\tau_1, \tau_2), (\tau_2, \tau_3), (\tau_3, \tau_4)\}$ is a query chain, as can be verified by evaluating its matrix representation.*

Let us denote as $\boldsymbol{r} = (r(s,a))_{(s,a) \in \mathcal{S} \times \mathcal{A}} \in \mathbb{R}^{SA}$ the vector representing a candidate reward function, and as $\boldsymbol{p} = (\mathbb{P}(\tau > \tau'))_{(\tau, \tau') \in \mathcal{Q}} \in [0,1]^{|\mathcal{Q}|}$ the vector of probabilities defined such that the $i$-th element of $\boldsymbol{p}$ refers to the $i$-th

---

query in $\mathcal{Q}$ and the $i$-th row of $\mathbf{Q}$. Since $\boldsymbol{p}$ is known, we can recover every $\boldsymbol{r} \in \mathcal{R}(\mathbb{P}, \sigma)$ by solving the following linear system of equations:

$$
\begin{bmatrix} \mathbf{Q} \\ \boldsymbol{e}_{SA}^{\top} \end{bmatrix} \boldsymbol{r} = \begin{bmatrix} \sigma^{-1}(\boldsymbol{p}) \\ x \end{bmatrix}, \tag{4}
$$

where $\sigma^{-1}(\boldsymbol{p})$ is the vector of the element-wise inverse sigmoid applied to the elements of $\boldsymbol{p}$, $\boldsymbol{e}_{SA}$ is the $SA$-th element of the canonical base, and $x \in \mathbb{R}$ accounts for the translation invariance of the likelihood function (for simplicity, we assume $x = 0$). It is worth noting that the matrix in the LHS of Equation (4) is full-rank and, thus, a unique solution exists. By repeating this for every stage $h \in [\![H]\!]$, we obtain the full reward function. This corresponds to exactly $H(SA - 1)$ queries.

## 5.2. Bernoulli Feedback

Consider now the realistic case in which, upon presenting a query $(\tau, \tau')$ to the human labeler, we receive as feedback a sample $y \sim \text{Ber}(\mathbb{P}(\tau > \tau'))$. The overall approach proceeds as in Section 5.1; however, here we do not have access to the true probability vector $\boldsymbol{p}$. Instead, by collecting multiple samples of each query, we can compute an estimate $\widehat{\boldsymbol{p}}$ of $\boldsymbol{p}$. Thus, we can substitute $\boldsymbol{p}$ with $\widehat{\boldsymbol{p}}$ in Equation (4), to recover a *reward function estimate* $\widehat{\boldsymbol{r}}$. Whereas with probability feedback every query chain $\mathcal{Q} \in \mathfrak{C}$ is equivalent in terms of reward function identification, here we have to account for the estimation error of each element in $\widehat{\boldsymbol{p}}$. Thus, we want to select the query chain that *injects* the least possible error in the estimated reward. To this end, we now introduce the notion of *index* of a query set.

**Definition 5.1** (Index of the Query Set). *Let $\mathcal{Q} \subseteq \mathcal{T} \times \mathcal{T}$ be a query set. We define its* index *$V(\mathcal{Q})$ as the* minimum variance *of the probability distributions of the queries it contains:*

$$
V(\mathcal{Q}) := \min_{(\tau, \tau') \in \mathcal{Q}} \left( \mathbb{P}(\tau > \tau') \right) \left( 1 - \mathbb{P}(\tau > \tau') \right).
$$

Recall that, under the sigmoid likelihood function, the probability of observing a preference increases with the difference in terms of utility between the two trajectories. Thus, the index of a query set $\mathcal{Q}$ is directly related to the query in $\mathcal{Q}$ with the maximum underlying utility gap, i.e., the minimum variance. The following result links the index of a query chain with the query complexity needed to identify an $\varepsilon$-compatible reward function, according to Definition 3.2.

**Lemma 5.2** (Query Complexity Upper Bound – Fixed Query Chain). *Let $\varepsilon \in [0, \sqrt{SAR_{\max}}]$ and $\delta \in (0,1)$. Let $\mathcal{Q} \in \mathfrak{C}$ be a query chain. A CRI-Tf algorithm that has knowledge of $V(\mathcal{Q})$ and presents every query to the human $\widetilde{O}(S^3 A^3 H \exp(\varepsilon(SA)^{-3/2}) \ln(1/\delta) \varepsilon^{-2} V(\mathcal{Q})^{-2})$ times is*

---

[3]This is a simplistic scenario, as it is unreasonable to expect a probability as feedback. The aim of this part is to guide the reader to a better understanding of the remainder of the paper by avoiding treating the statistical uncertainty.

---

**Algorithm 1:** $\lambda$-SQE.

**Input** : Query set $\mathcal{Q}$, parameters $\lambda \geqslant 0, \delta > 0$

1 **Initialize:** $\mathcal{X} \leftarrow \mathcal{Q}, \widehat{\boldsymbol{p}} \leftarrow \{0\}^{|\mathcal{Q}|}, t \leftarrow 1$
2 **repeat**
3     $\beta(t) \leftarrow \sqrt{\frac{\ln(4SA(SA-1)t^2/\delta)}{2t}}$,
4     $U \leftarrow \{0\}^{|\mathcal{X}|}, L \leftarrow \{0\}^{|\mathcal{X}|}$
5     **for** $i \in [\![|\mathcal{X}|]\!]$ **do**
6        Sample query $q_i$
7        Update preference estimator $p_i$
8        $U_i \leftarrow \max_{\widetilde{p} \in [p_i - \beta(t), p_i + \beta(t)]} \widetilde{p}(1 - \widetilde{p})$
9        $L_i \leftarrow \min_{\widetilde{p} \in [p_i - \beta(t), p_i + \beta(t)]} \widetilde{p}(1 - \widetilde{p})$
10     **end**
11     $I_L \leftarrow \texttt{BestBasis}(\mathcal{X}, L)$
12     **for** $i \in [\![|\mathcal{X}|]\!] : q_i \notin I_L$ **do**
13        $I_U(q_i) \leftarrow \texttt{BestBasis}(\mathcal{X}, U, q_i)$
14        **if** $\overline{V}(I_U(q_i)) < \underline{V}(I_L)$ **then**
15           $\mathcal{X} \leftarrow \mathcal{X} \backslash \{q_i\}, \widehat{\boldsymbol{p}} \leftarrow \widehat{\boldsymbol{p}} \backslash \{p_i\}$,
16           $U \leftarrow U \backslash \{U_i\}, L \leftarrow L \backslash \{L_i\}$
17        **end**
18     **end**
19     $I_L \leftarrow \texttt{BestBasis}(\mathcal{X}, L)$
20     $I_U \leftarrow \texttt{BestBasis}(\mathcal{X}, U)$
21     $t \leftarrow t + 1$
22 **until** $|\mathcal{X}| = SA - 1 \vee \overline{V}(I_U)/\underline{V}(I_L) \leqslant (1 + \lambda)$;

**Return :** Basis in $\mathcal{X}, \widehat{\boldsymbol{p}}, t$

---

**Algorithm 2:** BestBasis.

**Input** : Query set $\mathcal{Q}$, weights $\mathcal{W}$, [query $q_i$]

1 **if** $q_i$ is provided **then**
2     $w(q_i) \leftarrow \max_{q \in \mathcal{Q}} w(q) + 1$
3 **end**
4 Define graph
    $(\{\tau \in \mathcal{T} : \exists \tau' \in \mathcal{T} : (\tau, \tau') \in \mathcal{Q} \vee (\tau', \tau) \in \mathcal{Q}\}, \mathcal{Q})$
5 **for** $e \in \text{sorted}(\mathcal{Q}, \mathcal{W})$ **do**
6     **if** $\mathcal{Q} \backslash \{e\}$ is connected **then**
7        $\mathcal{Q} \leftarrow \mathcal{Q} \backslash \{e\}$
8     **end**
9 **end**

**Return :** Spanning Tree $\mathcal{Q}$

---

$(\varepsilon, \delta)$-*PAC with a query complexity of:*

$$\widetilde{O}\left(\frac{S^4 A^4 H \exp(\varepsilon(SA)^{-3/2}) \ln(1/\delta)}{\varepsilon^2 V(\mathcal{Q})^2}\right). \quad (5)$$

*Moreover, if* $R_{\max} \leqslant O(SA)$*, then the query sample complexity is upper bounded by:*

$$\widetilde{O}\left(\frac{S^4 A^4 H \ln(1/\delta)}{\varepsilon^2 V(\mathcal{Q})^2}\right). \quad (6)$$

Lemma 5.2 prescribes that the query complexity is minimized when the index is maximized, i.e., when $V(\mathcal{Q}) = 1/4$. This is due to the inverse sigmoid $\sigma^{-1}$, the derivative of which is minimized in $1/2$.

Thus, an efficient learner should devise a query chain whose index is as close as possible to $1/4$. We denote an *opti-*

*mal basis as* $\mathcal{Q}^* \in \arg\max_{\mathcal{Q} \in \mathfrak{B}} V(\mathcal{Q})$, and its index as $V_* = V(\mathcal{Q}^*)$. In particular, for a fixed stage $h \in [\![H]\!]$, it is intuitive to see that $\mathcal{Q}^*$ is a chain, obtained by reordering state-action pairs in terms of increasing rewards and devising a query set in which each state-action pair is compared to its successor. Ideally, an algorithm should employ $\mathcal{Q}^*$. However, since the probabilities $\mathbb{P}(\tau \succ \tau')$ are unknown, we cannot know $\mathcal{Q}^*$ before the learning process starts. Thus, *exploration* becomes necessary to find a *nearly-optimal* chain to reduce the query complexity. We represent near-optimality in a multiplicative sense w.r.t. the index of a query chain. In particular, let $\mathcal{X} \in \mathfrak{C}$, we say $\mathcal{X}$ is $\lambda$-optimal if $V_* \leqslant (1 + \lambda)V(\mathcal{X})$ for some $\lambda \geqslant 0$.

**Algorithm.** We now propose the $\lambda$-Successive Query Elimination ($\lambda$-SQE, Algorithm 1) procedure based on the elimination criteria of *Successive Elimination* (Even-Dar et al., 2006), identifying a $\lambda$-optimal chain w.p. at least $1 - \delta$, for $\lambda \geqslant 0$ and $\delta \in (0, 1)$.

First, let us introduce an auxiliary procedure employed by $\lambda$-SQE, namely BestBasis (Algorithm 2). Given a query set $\mathcal{Q} = \{q_i := (\tau_i, \tau_i')\}_{i=1}^{|\mathcal{Q}|}$ and a set of weights $\mathcal{W} = \{w(q_i)\}_{q_i \in \mathcal{Q}}$, denote the set of all trajectories in $\mathcal{Q}$ as $\mathcal{T}(\mathcal{Q}) := \{\tau \in \mathcal{T} : \exists \tau' \in \mathcal{T} : (\tau, \tau') \in \mathcal{Q} \vee (\tau', \tau) \in \mathcal{Q}\}$. BestBasis computes the minimum set of queries $\mathcal{Q}^\dagger \subseteq \mathcal{Q}$ with the *highest minimum weight*, i.e., $\mathcal{Q}^\dagger := \arg\max_{\mathcal{Q}' \subseteq \mathcal{Q}: \mathcal{T}(\mathcal{Q}') = \mathcal{T}(\mathcal{Q})} \min_{q \in \mathcal{Q}'} w(q)$. It constructs a graph in which trajectories are nodes and queries in $\mathcal{Q}$ are edges. Then, it iterates over queries by increasing weights, removing queries that do not partition the graph. Finally, it outputs a *spanning tree* (Diestel, 2012), i.e., a basis, with maximum minimum weight.

$\lambda$-SQE tackles the problem for each stage independently from the others, employing for each stage a query set $\mathcal{Q}^{(h)}$ defined as:

$$\mathcal{Q}^{(h)} = \Big\{ \left(\tau \overset{h}{\leftarrow} (s, a), \tau \overset{h}{\leftarrow} (s', a')\right),$$
$$\forall (s, a), (s', a') \in \mathcal{S} \times \mathcal{A} : (s, a) > (s', a')\Big\}, \quad (7)$$

where $>$ is an arbitrary lexicographic order over $\mathcal{S} \times \mathcal{A}$. $\lambda$-SQE iteratively discards queries that, with high probability, are not part of any $\lambda$-optimal basis. At each iteration, it adopts a round-robin procedure to present each query in the remaining set $\mathcal{X}$ to the human labeler, update the corresponding preference estimator $p_i$, and upper $U_i$ and lower $L_i$ confidence bounds of the variance of $p_i$. At Line 11, the procedure employs BestBasis to compute the *pessimistic* best basis $I_L$, i.e., the basis that maximizes the minimum lower confidence bound of the variance of its elements. $\lambda$-SQE then verifies if any query $q_i$ can be discarded. To do so, it first computes the *optimistic* best basis $I_U(q_i)$, using upper confidence bounds of the variances as

weights, and constraining $q_i$ to be part of the basis.[4] A query is discarded, i.e., Line 14, if the optimistic index $\overline{V}(I_U(q_i))$ with the query constraint is lower than the unconstrained pessimistic index $\underline{V}(I_L)$, meaning that $q_i$ is not contained in any $\lambda$-optimal basis with high probability. The optimistic and pessimistic indices are computed as follows: $\overline{V}(\mathcal{Z}) := \min_{p \in U_{\mathcal{Z}}} p(1-p)$ and $\underline{V}(\mathcal{Z}) := \min_{p \in L_{\mathcal{Z}}} p(1-p)$, where $U_{\mathcal{Z}} = \{U_i \in U : q_i \in \mathcal{Z} \; \forall i \in [\![|\mathcal{X}|]\!]\}$, and $L_{\mathcal{Z}} = \{L_i \in L : q_i \in \mathcal{Z} \; \forall i \in [\![|\mathcal{X}|]\!]\}$. $\lambda$-SQE terminates when either a single query basis remains or the ratio between the unconstrained optimistic index of the best basis w.r.t. $U$ and the pessimistic index of the best basis w.r.t. $L$ is lower than $1 + \lambda$. Finally, it returns a query basis (in particular, a chain), its probability estimators, and the termination round.

For ease of presentation, we define the following notion.

**Definition 5.2** (Query Suboptimality Gap). *Let* $q := (\tau \xleftarrow{h} (s,a), \tau \xleftarrow{h} (s',a'))$ *be a query, with* $h \in [\![H]\!]$, *and let* $V_*$ *be the index of the optimal basis. The* suboptimality gap *of query* $q$ *is defined as* $\Delta_q := V_* - \max_{I \in \mathfrak{B}:q \in I} V_I$.

We now provide a result on the query complexity of discarding a query, based on its suboptimality gap.

**Lemma 5.3.** *Let* $\lambda$-SQE *run with query set* $\mathcal{Q}^{(h)}$ *and parameters* $\lambda \geqslant 0$ *and* $\delta \in (0,1)$. *Let* $q \in \mathcal{Q}^{(h)}$ *such that* $\Delta_q > 0$. *Then,* $\lambda$-SQE *discards query* $q$ *after a number of rounds that is bounded by* $\widetilde{O}(\ln(1/\delta)/\Delta_q^2)$.

Ideally, we want to set $\lambda = 0$, making $\lambda$-SQE terminate when only the optimal chain remains, w.p. at least $1 - \delta$. We provide an upper bound on its query complexity.

**Theorem 5.4** (Exact Optimal Basis Identification). *Let* $\mathcal{Q}^{(h)}$ *be a query set,* $\lambda = 0$, *and* $\delta \in (0,1)$. $\lambda$-SQE *returns the optimal chain* $\mathcal{Q}^*$ *w.p. at least* $1 - \delta$ *with query complexity bounded by:*

$$\widetilde{O}\left( \frac{(SA-1)\ln(1/\delta)}{\Delta_{SA}^2} + \sum_{i=SA}^{|\mathcal{Q}^{(h)}|} \frac{\ln(1/\delta)}{\Delta_i^2} \right),$$

*where* $i > j \Rightarrow \Delta_i > \Delta_j$.

Clearly, the query complexity grows indefinitely if there exists a single query $q \in \mathcal{Q}^{(h)}$ such that $\Delta_q \to 0$. Thus, we accept to settle for a *nearly*-optimal query chain in exchange for a drastically reduced query complexity.

**Theorem 5.5** ($\lambda$-Optimal Basis Identification). *Let* $\mathcal{Q}^{(h)}$ *be a query set,* $\lambda > 0$, *and* $\delta \in (0,1)$. $\lambda$-SQE *returns a* $\lambda$-*optimal basis w.p. at least* $1 - \delta$ *with query complexity bounded by:*

$$\widetilde{O}\left( \sum_{q \in \mathcal{Q}^{(h)}} \min\left\{ \frac{\ln(1/\delta)}{\Delta_q^2}, \frac{(2+\lambda)^2 \ln(1/\delta)}{\lambda^2 V_*^2} \right\} \right). \quad (8)$$

As discussed earlier in this section, the query complexity is

minimized when the index of the query basis is maximized, and $V_*$ corresponds to the maximum minimum variance of any query in a basis. The idea that the query complexity is minimized when the index equals $1/4$ may seem counterintuitive under the BT model. Ideally, with the objective of minimizing query complexity, we would expect an algorithm to select queries with larger gaps in the underlying trajectory returns, as they, informally, *yield more information about human preferences*. What we can extrapolate from previous results, instead, is quite the opposite, as we should select pairs of trajectories that are as similar as possible. This may be an implicit limitation of the BT model when applied to the compatible reward identification task.

For completeness' sake, in Appendix C we report an algorithm which employs the query chain returned by $\lambda$-SQE to identify a compatible reward function, with an overall query complexity bounded by $\widetilde{O}(S^4 A^4 H \ln(1/\delta)/(\varepsilon^2 V_*^2))$, subject to a convenient selection of $\lambda$, making its query complexity of the same order as an algorithm that knows the optimal basis and its index prior to the learning process.

## 6. Conclusions

In this paper, we have deepened the understanding of the problem of reconstructing the set of feasible rewards under preference feedback. In particular, we have formulated the problem as finding the set of queries, i.e., comparisons between trajectories, to be posed to the user in order to accurately recover the set of rewards with high probability. Based on the observation that the practical cost corresponds to the number of times the human annotator is asked, we have focused on studying the query complexity. First, we discussed that an exponential complexity in the horizon $H$ is unavoidable when considering non-Markovian utilities, leading us to consider (possibly surrogate) Markovian rewards. Then, when preferences are delivered deterministically, we provided lower bounds on the query complexity and proved that using fewer queries than prescribed can lead to arbitrarily large identification errors. Finally, moving to stochastic preferences generated by the BT model, we have introduced the notion of query index to characterize how much a query set can accelerate the identification of feasible rewards. This analysis has highlighted that queries associated with mild preferences (i.e., where the preference probability is close to $1/2$) are more effective, revealing a potential drawback of the BT model.

**Future Work.** A limitation of the proposed approaches is that they assume the ability to construct trajectories at will, without accounting for the sample cost required to generate such trajectories in the environment. Future work should focus on the case where the environment can be explored only through interaction using an exploration policy.

---

[4]This is achievable by setting $w(q_i) = \max_{q \in \mathcal{X}} w(q) + 1$.

## Acknowledgements

This publication was funded with the contribution of Ministero dell'Università e della Ricerca pursuant to D.D. n. 7206 of 17 April 2025 – BANDO FIS 2. Project FIS-2023-02598 (Starting Grant), title: "Unified Learning from Diverse Human Feedback" (HUmLrn). CUP: D53C25000710001.

## Impact Statement

This paper presents work whose goal is to advance the field of machine learning. There are many potential societal consequences of our work, none of which we feel must be specifically highlighted here.

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

# A. Omitted Proofs

In this appendix, we report the proofs omitted in the main paper.

## A.1. Proof of Theorem 4.3

Given $\mathcal{S}$, $\mathcal{A}$, and $H$, for every $r \in \mathfrak{R}$ it is possible to devise a query set $\mathcal{Q}$ such that it is minimal w.r.t. the identification of $r$ under the likelihood function $g = \mathbb{H}$. By Definition 3.1, $\mathcal{R}(\mathbb{P}, \mathbb{H}; \mathcal{Q}) = \mathcal{R}(\mathbb{P}, \mathbb{H})$, and there exists no strict subset $\mathcal{Q}' \subset \mathcal{Q}$ such that $\mathcal{R}(\mathbb{P}, \mathbb{H}; \mathcal{Q}') = \mathcal{R}(\mathbb{P}, \mathbb{H})$.

Clearly, $\mathcal{H}(\mathcal{R}(\mathbb{P}, \mathbb{H}; \mathcal{Q}), \mathcal{R}(\mathbb{P}, \mathbb{H})) = 0$. Moreover, the selection of function $g$ allows the reconstruction on a *total order relation* among trajectories, and $\mathcal{R}(\mathbb{P}, \mathbb{H})$ corresponds to every reward function that induces such a total ordering.

Now, considering a query $(\tau, \tau') \in \mathcal{Q}$, it follows by the minimality of $\mathcal{Q}$ that $\mathcal{Q} \backslash \{(\tau, \tau')\}$ is not sufficient. This, in turn, means that there exists an MDP such that, by asking queries in $\mathcal{Q} \backslash \{(\tau, \tau')\}$, it is impossible to infer whether $\tau > \tau'$ or $\tau' > \tau$. In particular, this can happen when $\tau$ and $\tau'$ are one the direct successor of the other in the total order relation induced by the underlying reward function.

It follows that $\mathcal{R}(\mathbb{P}, \mathbb{H}) \subset \mathcal{R}(\mathbb{P}, \mathbb{H}; \mathcal{Q} \backslash \{(\tau, \tau')\})$. In particular, $\mathcal{R}(\mathbb{P}, \mathbb{H}; \mathcal{Q} \backslash \{(\tau, \tau')\})$ can be partitioned into two subsets, considering that any reward function it contains must either induce $\tau > \tau'$ or $\tau' > \tau$. Obviously, one of the two partitions perfectly coincides with $\mathcal{R}(\mathbb{P}, \mathbb{H})$, whereas the other is disjoint from it. Denoting $\mathcal{R}_1 := \mathcal{R}(\mathbb{P}, \mathbb{H})$ and $\mathcal{R}_2 := \mathcal{R}(\mathbb{P}, \mathbb{H}; \mathcal{Q} \backslash \{(\tau, \tau')\}) \backslash \mathcal{R}(\mathbb{P}, \mathbb{H})$, we can then rewrite:

$$\mathcal{H}(\mathcal{R}(\mathbb{P}, \mathbb{H}; \mathcal{Q} \backslash \{(\tau, \tau')\}), \mathcal{R}(\mathbb{P}, \mathbb{H})) = \mathcal{H}(\mathcal{R}_1, \mathcal{R}_2). \tag{9}$$

Clearly, there exists an MDP in which $\tau$ and $\tau'$ are the direct successors of each other in the total order relation induced by the underlying reward function, and it is possible to define the reward functions $r_1 \in \mathcal{R}_1$ and $r_2 \in \mathcal{R}_2$ such that:

$$r_1(\tau) = \varepsilon, \qquad r_1(\tau') = R_{\max} - \varepsilon, \tag{10}$$
$$r_2(\tau) = R_{\max} - \varepsilon, \qquad r_2(\tau') = \varepsilon, \tag{11}$$

where $\varepsilon$ is employed to define the remaining rewards, simply to ensure that a total order relation can be induced among elements, finally, we get that:

$$\mathcal{H}(\mathcal{R}_1, \mathcal{R}_2) = R_{\max} - 2\varepsilon, \tag{12}$$

which tends to $R_{\max}$ as $\varepsilon \to 0$, thus concluding the proof.

## A.2. Proof of Lemma 5.2

Let a $(\varepsilon, \delta)$-PAC CRI-Tf algorithm employ query chain $\mathcal{Q} \in \mathfrak{C}$ as its *offline*, i.e., fixed, query set and fix a stage $h \in [\![H]\!]$. As stated in Definition 3.2, the goal of the algorithm is to ensure that:

$$\Pr(\mathcal{H}(\mathcal{R}(\mathbb{P}, \sigma), \mathcal{R}(\mathbb{P}, \sigma; \mathcal{Q})) \geqslant \varepsilon) \leqslant \delta, \tag{13}$$

while minimizing the query complexity, i.e., the number of interactions with the human.

Recall that the ternary matrix $\mathbf{Q}$ associated to $\mathcal{Q}$ is an upper bidiagonal matrix with $1$ on the main diagonal, $-1$ directly above the main diagonal, and $0$ everywhere else. Moreover, under the BT model, there exists a preference vector $\boldsymbol{p}$ associated with $\mathcal{Q}$ such that the $i$-th element of $\boldsymbol{p}$ represents the probability of the human providing feedback $1$ when presented with the $i$-th query in $\mathcal{Q}$. By fixing one element of the reward function to remove the translation invariance of the BT model, we recall that, were $\boldsymbol{p}$ known to the learner, a reward function $\boldsymbol{r} \in \mathcal{R}(\mathbb{P}, \sigma)$ could be recovered by solving the linear system of equations shown in Equation (4).

Given that $\boldsymbol{p}$ is unknown to the learner, the algorithm must present each query to the human a sufficient number of times to estimate a vector of preferences $\widehat{\boldsymbol{p}}$ and employ such a vector to recover a reward estimate $\widehat{\boldsymbol{r}}$ by solving a linear system of equations similar to that of Equation (4).

Thus, we can define a goal that ensures the condition in Equation (13) as:

$$\|\boldsymbol{r} - \widehat{\boldsymbol{r}}\|_2 \leqslant \varepsilon. \tag{14}$$

From Equation (4) we can derive:

$$\|r - \widehat{r}\|_2 = \left\| \begin{bmatrix} \mathbf{Q} \\ e_{SA}^\top \end{bmatrix}^{-1} \begin{bmatrix} \sigma^{-1}(p) - \sigma^{-1}(\widehat{p}) \\ 0 \end{bmatrix} \right\|_2 \tag{15}$$

$$\leqslant \left\| \begin{bmatrix} \mathbf{Q} \\ e_{SA}^\top \end{bmatrix}^{-1} \right\|_2 \left\| \begin{bmatrix} \sigma^{-1}(p) - \sigma^{-1}(\widehat{p}) \\ 0 \end{bmatrix} \right\|_2 \tag{16}$$

$$\leqslant \sigma_{\min}\left( \begin{bmatrix} \mathbf{Q} \\ e_{SA}^\top \end{bmatrix} \right)^{-1} \left\| \sigma^{-1}(p) - \sigma^{-1}(\widehat{p}) \right\|_2, \tag{17}$$

where Equation (16) follows from the Cauchy-Schwarz inequality and Equation (17) follows from observing that element $0$ has no impact on the L2-norm and that $\|\mathbf{A}^{-1}\|_2 = \sigma_{\min}(\mathbf{A})^{-1}$, where $\sigma_{\min}(\mathbf{A})$ represents the minimum singular value of a matrix $\mathbf{A}$.

Recalling that $\sigma_{\min}(\mathbf{A}) = \sqrt{\lambda_{\min}(\mathbf{A}\mathbf{A}^\top)}$, where $\lambda_{\min}(\mathbf{A})$ represents the minimum eigenvalue of matrix $\mathbf{A}$, we observe that:

$$\begin{bmatrix} \mathbf{Q} \\ e_{SA}^\top \end{bmatrix} \begin{bmatrix} \mathbf{Q} \\ e_{SA}^\top \end{bmatrix}^\top \tag{18}$$

is a *tridiagonal* matrix with element $2$ on the main diagonal, element $-1$ directly above and below the main diagonal, and $0$ everywhere else. The eigenvalues of such a matrix have been proved (Elliott, 1953; Kouachi, 2006) to be in the form:

$$\lambda_i = 2 - 2\cos\left( \frac{i\pi}{SA} \right), \tag{19}$$

for $i \in [\![SA]\!]$. As such, we can compute the minimum singular value as:

$$\sigma_{\min}\left( \begin{bmatrix} \mathbf{Q} \\ e_{SA}^\top \end{bmatrix} \right) = \sqrt{\min_{i \in [\![SA]\!]} 2 - 2\cos\left( \frac{i\pi}{SA} \right)} \tag{20}$$

$$= \min_{i \in [\![SA]\!]} \sqrt{2 - 2\cos\left( \frac{i\pi}{SA} \right)} \tag{21}$$

$$= \min_{i \in [\![SA]\!]} 2\sqrt{\frac{1 - \cos\left( \frac{i\pi}{SA} \right)}{2}} \tag{22}$$

$$= \min_{i \in [\![SA]\!]} 2\sin\left( \frac{i\pi}{2SA} \right) \tag{23}$$

$$\geqslant 2\sin\left( \frac{\pi}{2SA} \right) \tag{24}$$

$$\geqslant \frac{\pi}{2SA}, \tag{25}$$

where Equation (21) follows from the monotonicity of the square root, and Equation (23) follows from the half-angle formula for the sine.

Then, we define $w = \arg\max_{i \in [\![SA]\!]}(\sigma^{-1}(p_i) - \sigma^{-1}(\widehat{p}_i))$, and denote as $q = p_w$ and $\widehat{q} = \widehat{p}_w$ as the element of the vector of preference and corresponding estimator that maximize their difference after applying the inverse sigmoid, and we upper bound:

$$\left\| \sigma^{-1}(p) - \sigma^{-1}(\widehat{p}) \right\|_2 \leqslant \sqrt{(SA - 1)(\sigma^{-1}(q) - \sigma^{-1}(\widehat{q}))^2}. \tag{26}$$

By combining Equations (25) and (26) into Equation (17), and setting it to be less than or equal to $\varepsilon$, we get:

$$\frac{2}{\pi} SA\sqrt{(SA - 1)(\sigma^{-1}(q) - \sigma^{-1}(\widehat{q}))^2} \leqslant \varepsilon. \tag{27}$$

By rearranging terms, we obtain:

$$|\sigma^{-1}(q) - \sigma^{-1}(\widehat{q})| \leqslant \frac{\pi\varepsilon}{2SA\sqrt{SA - 1}} =: \overline{\varepsilon}. \tag{28}$$

By defining the estimator as $\hat{q} = q + \Delta q$ and rearranging terms, we can write:

$$\sigma\left(\sigma^{-1}(q) - \bar{\varepsilon}\right) - q \leqslant \Delta q \leqslant \sigma\left(\sigma^{-1}(q) + \bar{\varepsilon}\right) - q. \tag{29}$$

By recalling that $\sigma(x) = 1/(1 + e^{-x})$ and $\sigma^{-1}(x) = \ln(x/(1-x))$, we can rewrite Equation (29) as:

$$\underbrace{q\left(\frac{1}{q + (1-q)e^{\bar{\varepsilon}}} - 1\right)}_{(a)} \leqslant \Delta q \leqslant \underbrace{q\left(\frac{1}{q + (1-q)e^{-\bar{\varepsilon}}} - 1\right)}_{(b)}. \tag{30}$$

By ensuring that the estimation error of the probability element $q$ lies within the interval in Equation (30), we ensure that Equation (14) holds. However, such an interval is asymmetric. To allow the extrapolation of a sample complexity, we first make the interval symmetric by upper-bounding term $(a)$ and lower-bounding term $(b)$, thus getting a stricter condition on the estimation error.

By first-order Taylor series expansion in $\bar{\varepsilon} = 0$, we get:

$$(a) \leqslant \max_{\xi \in [0, \bar{\varepsilon}]}\left(-\frac{q(1-q)e^{\xi}}{(q + (1-q)e^{\xi})^2}\right)\bar{\varepsilon} \tag{31}$$

$$\leqslant \max\left\{-1, -\frac{e^{\bar{\varepsilon}}}{(q + (1-q)e^{\bar{\varepsilon}})^2}\right\}q(1-q)\bar{\varepsilon} \tag{32}$$

$$\leqslant -\min\left\{1, e^{-\bar{\varepsilon}}\right\}q(1-q)\bar{\varepsilon} \tag{33}$$

$$\leqslant -q(1-q)e^{-\bar{\varepsilon}}\bar{\varepsilon}, \tag{34}$$

where Equation (32) follows by observing that the maximum of the function inside the $\max$ operator will lie in one of the edges of the interval, due to *(i)* it being negative for any value $q \in [0, 1]$ and $\xi \in \mathbb{R}$, *(ii)* having limit $0^-$ for $\xi \to \pm\infty$, and *(iii)* having two changes in concavity, thus having its stationary point in the minimum. Equation (33) follows from observing that $q + (1-q)e^{\bar{\varepsilon}} \leqslant e^{\bar{\varepsilon}}$, for $\bar{\varepsilon} > 0$.

In a similar manner, we can derive:

$$(b) \geqslant \min_{\xi \in [0, \bar{\varepsilon}]}\left(\frac{q(1-q)e^{\xi}}{(1 + q(e^{\xi} - 1))^2}\right)\bar{\varepsilon} \tag{35}$$

$$\geqslant \min\left\{1, \frac{e^{\bar{\varepsilon}}}{(1 + q(e^{\bar{\varepsilon}} - 1))^2}\right\}q(1-q)\bar{\varepsilon} \tag{36}$$

$$\geqslant \min\{1, e^{-\bar{\varepsilon}}\}q(1-q)\bar{\varepsilon} \tag{37}$$

$$\geqslant q(1-q)e^{-\bar{\varepsilon}}\bar{\varepsilon}, \tag{38}$$

where Equation (36) follows by observing that the function inside the $\min$ operator is concave and thus its minimum is in one of the edges of the minimization interval, and Equation (37) follows from observing that $1 + q(e^{\bar{\varepsilon}} - 1) \leqslant e^{\bar{\varepsilon}}$. By combining Equations (34) and (38) into Equation (30), we get the symmetric interval:

$$-q(1-q)e^{-\bar{\varepsilon}}\varepsilon \leqslant \Delta q \leqslant q(1-q)e^{-\bar{\varepsilon}}\varepsilon. \tag{39}$$

By the Chernoff-Hoeffding inequality, we can derive that, with a union bound on all queries in $\mathcal{Q}$ and rounds, the minimum number of samples, i.e., rounds, required to ensure that the estimation error of probability $q$ lies in the interval shown in Equation (39) is:

$$t \geqslant \frac{\ln\left(\frac{8(SA-1)t^2}{\delta}\right)}{2\Delta q^2}. \tag{40}$$

By following Lemma 12 of (Jonsson et al., 2020), we can derive that, by selecting:

$$t \geqslant \frac{2e^{2\bar{\varepsilon}}}{\bar{\varepsilon}^2(q(1-q))^2}\ln\left(\frac{3e^{2\bar{\varepsilon}}SA}{\bar{\varepsilon}^2(q(1-q))^2\delta}\right), \tag{41}$$

we guarantee that the error in the estimation of probability $q$ is less than $\bar{\varepsilon}$ w.p. at least $1 - \delta$. Given the use of the

Chernoff-Hoeffding inequality and the uniform number of samples for the queries, we observe that the element $q \in \boldsymbol{p}$ can be replaced with the element farthest from $1/2$. Thus, we can rewrite:

$$t \geqslant \frac{2e^{2\bar{\varepsilon}}}{\bar{\varepsilon}^2 V(\mathcal{Q})^2} \ln\left(\frac{3e^{2\bar{\varepsilon}}SA}{\bar{\varepsilon}^2 V(\mathcal{Q})^2\delta}\right), \tag{42}$$

where $V(\mathcal{Q})$ is the index of the query set as per Definition 5.1. By replacing the definition of $\bar{\varepsilon}$, we then get:

$$t \geqslant \frac{8S^2A^2(SA-1)\exp\left(\frac{\pi\varepsilon}{SA\sqrt{SA-1}}\right)}{\pi^2\varepsilon^2 V(\mathcal{Q})^2} \ln\left(\frac{12S^2A^2(SA-1)\exp\left(\frac{\pi\varepsilon}{SA\sqrt{SA-1}}\right)}{\pi^2\varepsilon^2 V(\mathcal{Q})^2\delta}\right).$$

Moreover, considering that the range of values of $\varepsilon$ is $[0, \sqrt{SA}R_{\max}]$, it follows that, if $R_{\max} \leqslant (S^2A^2)/\pi$, we obtain:

$$t \geqslant \frac{8eS^2A^2(SA-1)}{\pi^2\varepsilon^2 V(\mathcal{Q})^2} \ln\left(\frac{12eS^2A^2(SA-1)}{\pi^2\varepsilon^2 V(\mathcal{Q})^2\delta}\right). \tag{43}$$

Multiplying these results by $SA - 1$, i.e., the number of queries to sample, we get the query complexity of a single stage. By summing over all stages, we conclude the proof. In general, the value of $V(\mathcal{Q})$ may differ between stages; however, we consider it constant across stages for ease of presentation.

### A.3. Proof of Lemma 5.3

Let $q \in \mathcal{Q}^{(h)}$ be a suboptimal query, i.e., $\Delta_q > 0$. We want to bound the number of rounds after which, with high probability, $q$ is identified as a suboptimal query and thus discarded. To improve the readability, let us restate the discard condition of a suboptimal query, i.e., Line 14 of Algorithm 1: a suboptimal query $q \in \mathcal{X} \subseteq \mathcal{Q}^{(h)}$ is discarded at round $t$ if the following condition holds:

$$\overline{V}(I_U(q)) < \underline{V}(I_L), \tag{44}$$

where $I_U(q)$ is the *optimistic best basis* computed in terms of the upper confidence bounds of the variance of the probabilities and constraining $q$ to be an element of the basis, $I_L$ is the *pessimistic best basis* computed in terms of the lower confidence bounds of the variance of the probabilities, $\overline{V}(I_U(q)) = \min_{p\in U_{I_U}(t)} p(1-p)$ is the optimistic index at round $t$, and $\underline{V}(I_L) = \min_{p\in L_{I_L}(t)} p(1-p)$ is the pessimistic index at round $t$. Combining Equation (44) with the definitions of the elements that compose it, we can rewrite the discard condition as:

$$\max_{I_U\in\mathfrak{B}(\mathcal{X}):q\in I_U} \min_{j\in I_U} U_j(t) < \max_{I_L\in\mathfrak{B}(\mathcal{X})} \min_{j\in I_L} L_j(t), \tag{45}$$

where $\mathfrak{B}(\mathcal{X})$ is the set of bases that can be constructed with queries in $\mathcal{X}$ and $U_j(t)$ (resp. $L_j(t)$) represents the element of $U$ (resp. $L$) that corresponds to query $j$, computed after $t$ rounds.

Let us first introduce some auxiliary lemmas that will be required in the analysis of the $\lambda$-SQE algorithm. First, we provide the definition of the *good event*, and demonstrate it holds with high probability.

**Lemma A.1** (Good Event). *Fix $\delta > 0$. Let $\mathcal{Q}^{(h)}$ be a query set, $\boldsymbol{p}$ be its associated probability vector, $\widehat{\boldsymbol{p}}$ its estimator, and let $\widehat{p}_i(t)$ be the estimator of the $i$-th element of $\boldsymbol{p}$ after $t > 0$ samples, i.e., rounds. Let $\mathcal{E}$ be the event defined as:*

$$\mathcal{E} := \left\{ |\widehat{p}_i(t)(1 - \widehat{p}_i(t)) - p_i(1 - p_i)| \leqslant \sqrt{\frac{\ln\left(\frac{4SA(SA-1)t^2}{\delta}\right)}{2t}}, \forall i \in [\![|\mathcal{Q}^{(h)}|]\!], t > 0 \right\}. \tag{46}$$

*Then, event $\mathcal{E}$ holds w.p. at least $1 - \delta$.*

*Proof.* Fix $\delta > 0$, $i \in [\![|\mathcal{Q}^{(h)}|]\!]$, and $t > 0$. Then, by the Chernoff-Hoeffding inequality, w.p. at least $1 - \delta$ it holds that:

$$|p_i(t) - \widehat{p}_i(t)| \leqslant \sqrt{\frac{\ln(2/\delta)}{2t}}. \tag{47}$$

By observing that $|\mathcal{Q}^{(h)}| \leqslant SA(SA-1)/2$ and by applying a union bound over all queries and rounds, we get that:

$$|p_i(t) - \widehat{p}_i(t)| \leqslant \sqrt{\frac{\ln\left(\frac{4SA(SA-1)t^2}{\delta}\right)}{2t}}, \tag{48}$$

holds simultaneously for all queries and rounds w.p. at least $1 - \delta$ since:

$$2\frac{SA(SA-1)}{2}\sum_{t=1}^{\infty}\frac{\delta}{4SA(SA-1)t^2} = \frac{\delta}{4}\sum_{t=1}^{\infty}t^{-2} = \frac{\delta}{4}\frac{\pi^2}{6} \leqslant \delta \tag{49}$$

By defining the estimator as $\widehat{p}_i(t) = p_i + \Delta p_i(t)$, we now provide a concentration bound on the variance of the estimator:

$$|\widehat{p}_i(t)(1 - \widehat{p}_i(t)) - p_i(1 - p_i)| = |(p_i + \Delta p_i(t))(1 - p_i - \Delta p_i(t)) - p_i(1 - p_i)| \tag{50}$$
$$= |\Delta p_i(t)(1 - p_i - \widehat{p}_i(t))| \tag{51}$$
$$\leqslant |\Delta p_i(t)|, \tag{52}$$

where Equation (52) follows by observing that $(1 - p_i - \widehat{p}_i(t)) \in [-1, 1]$. Finally, we derive that:

$$|\widehat{p}_i(t)(1 - \widehat{p}_i(t)) - p_i(1 - p_i)| = |\Delta p_i(t)| \leqslant \sqrt{\frac{\ln\left(\frac{4SA(SA-1)t^2}{\delta}\right)}{2t}}, \tag{53}$$

holds simultaneously for all queries and rounds w.p. at least $1 - \delta$, thus concluding the proof. $\square$

Then, we demonstrate that, under the good event, the optimistic and pessimistic indices of a basis are an upper and a lower bound, respectively, of the true index of the basis.

**Lemma A.2.** *Let event $\mathcal{E}$ hold. Then, for every basis $I \in \mathfrak{B}(\mathcal{X})$ and $t > 0$, it holds that:*

$$\underline{V}(I) \leqslant V(I) \leqslant \overline{V}(I). \tag{54}$$

*Proof.* The proof trivially follows by observing that, under event $\mathcal{E}$, the following equations hold:

$$\underline{V}(I) = \min_{j \in I} L_j(t) \leqslant \min_{j \in I} p_j(1 - p_j) = V(I), \tag{55}$$
$$\overline{V}(I) = \min_{j \in I} U_j(t) \geqslant \min_{j \in I} p_j(1 - p_j) = V(I), \tag{56}$$

where $p_j$ denotes the underlying probability of observing feedback 1 associated to query $j \in I$. $\square$

Lastly, we demonstrate that, under the good event, the optimal basis is never discarded. We say that *a basis is discarded* if any of its elements is discarded.

**Lemma A.3.** *Let event $\mathcal{E}$ hold, and let $\mathcal{Q}^*$ be an optimal basis of $\mathcal{Q}$. Then, for every round $t > 0$, $\mathcal{Q}^*$ is never discarded, i.e., for every $i^* \in \mathcal{Q}^*$, it holds that:*

$$\max_{I \in \mathfrak{B}(\mathcal{X}):i^* \in I} \min_{j \in I} U_j(t) \geqslant \max_{I \in \mathfrak{B}(\mathcal{X})} \min_{j \in I} L_j(t). \tag{57}$$

*Proof.* We demonstrate this result by contradiction. Let event $\mathcal{E}$ hold and let $i^* \in \mathcal{Q}^*$. If $i^*$ is discarded, then there exists a round $t > 0$ such that:

$$\max_{I \in \mathfrak{B}(\mathcal{X}):i^* \in I} \min_{j \in I} U_j(t) < \max_{I \in \mathfrak{B}(\mathcal{X})} \min_{j \in I} L_j(t). \tag{58}$$

By Lemma A.2, it holds that:

$$\max_{I \in \mathfrak{B}(\mathcal{X}):i^* \in I} \min_{j \in I} U_j(t) \geqslant \max_{I \in \mathfrak{B}(\mathcal{X}):i^* \in I} V(I), \tag{59}$$

and:

$$\max_{I \in \mathfrak{B}(\mathcal{X})} \min_{j \in I} L_j(t) \leqslant \max_{I \in \mathfrak{B}(\mathcal{X})} V(I). \tag{60}$$

By combining the two into Equation (58), we derive that:

$$\max_{I \in \mathfrak{B}(\mathcal{X}):i^* \in I} V(I) < \max_{I \in \mathfrak{B}(\mathcal{X})} V(I), \tag{61}$$

which, since $i^* \in \mathcal{Q}^*$ by definition, corresponds to:

$$V_* < V_*, \tag{62}$$

which is a contradiction, thus concluding the proof. $\qquad\square$

We can now address the proof of Lemma 5.3. Recalling the discard condition of a suboptimal query, i.e., Equation (45), and observing that:

$$\max_{I_L \in \mathfrak{B}(\mathcal{X})} \min_{j \in I_L} L_j(t) \geq \min_{j \in \mathcal{Q}^*} L_j(t), \tag{63}$$

we can rewrite a stricter discard condition to be verified as:

$$\max_{I_U \in \mathfrak{B}(\mathcal{X}):q \in I_U} \min_{j \in I_U} U_j(t) < \min_{j \in \mathcal{Q}^*} L_j(t). \tag{64}$$

Let us denote as $\beta(t) := \sqrt{\frac{\ln(4SA(SA-1)t^2/\delta)}{2t}}$ the Chernoff-Hoeffding confidence bound at round $t$ and as $\widehat{V}_i(t) := \widehat{p}_i(1-\widehat{p}_i)$ the empirical variance of a query $i$. We now lower bound the RHS of Equation (64) as:

$$\min_{j \in \mathcal{Q}^*} L_j(t) \geq \min_{j \in \mathcal{Q}^*} \widehat{V}_j(t) - \beta(t) \tag{65}$$

$$\geq V_* - 2\beta(t), \tag{66}$$

where Equation (65) follows from observing that the confidence intervals derived via the Chernoff-Hoeffding inequality depend only on the number of rounds and that the queries in $\mathcal{X}$ are sampled in a round robin manner, and by the definition of $L_j(t)$ for any query. Equation (66) follows by observing that, by Lemma A.1, the estimated variance of any query is at most at a distance $\beta(t)$ from the true variance. Similarly, we can upper bound the LHS of Equation (64) as:

$$\max_{I_U \in \mathfrak{B}(\mathcal{X}):q \in I_U} \min_{j \in I_U} U_j(t) \leq \max_{I_U \in \mathfrak{B}(\mathcal{X}):q \in I_U} \min_{j \in I_U} \widehat{V}_j(t) + \beta(t) \tag{67}$$

$$\leq \max_{I_U \in \mathfrak{B}(\mathcal{X}):q \in I_U} V(I_U) + 2\beta(t). \tag{68}$$

By combining Equations (66) and (68) into Equation (64), we derive:

$$\max_{I_U \in \mathfrak{B}(\mathcal{X}):q \in I_U} V(I_U) + 2\beta(t) < V_* - 2\beta(t). \tag{69}$$

By isolating $\beta(t)$ and applying Definition 5.2, we obtain:

$$\frac{\Delta_q}{4} > \beta(t) = \sqrt{\frac{\ln\left(\frac{4SA(SA-1)t^2}{\delta}\right)}{2t}}. \tag{70}$$

By rearranging terms, we get:

$$\Delta_q^2 t > 16 \ln\left(\frac{2SAt}{\delta}\right). \tag{71}$$

Again, following Lemma 12 of (Jonsson et al., 2020), we derive:

$$t > \frac{32}{\Delta_q^2} \ln\left(\frac{32SA}{\Delta_q^2 \delta}\right), \tag{72}$$

thus concluding the proof.

### A.4. Proof of Theorem 5.4

By Lemma A.1, event $\mathcal{E}$ holds w.p. at least $1 - \delta$. Let us sort the queries in terms of increasing suboptimality gap, with the $SA - 1$ queries in $\mathcal{Q}^*$, the suboptimality gap of which is $0$, occupy the first $SA - 1$ positions after sorting.

The result of the Theorem then simply follows by summing the query complexity of each suboptimal query, and observing that each query in $\mathcal{Q}^*$ is sampled the same number of times as the first suboptimal query. Thus, we derive that, with an overall number of interactions with the human that is at least:

$$t \geqslant \frac{32(SA - 1)\ln\left(\frac{32SA}{\Delta_q^2 \delta}\right)}{\Delta_{SA}} + \sum_{i=SA}^{|\mathcal{Q}^{(h)}|} \frac{32\ln\left(\frac{32SA}{\Delta_i^2 \delta}\right)}{\Delta_i^2}, \tag{73}$$

the optimal basis is identified w.p. at least $1 - \delta$.

### A.5. Proof of Theorem 5.5

Fix a stage $h \in [\![H]\!]$, and let $\mathcal{Q}^{(h)}$ be a query set, $\lambda > 0$, and $\delta > 0$. In order to bound the query complexity of identifying a $\lambda$-optimal basis w.p. at least $1 - \delta$, we first need to bound the number of rounds after which the stopping condition at Line 22 of Algorithm 1.

Recalling that $\lambda$-SQE halts execution either when a single basis remains, or when:

$$\frac{\overline{V}_{I_U}}{\underline{V}_{I_L}} = \frac{\max_{I_U \in \mathfrak{B}(\mathcal{X})} \min_{j \in I_U} U_j(t_{\text{stop}})}{\max_{I_L \in \mathfrak{B}(\mathcal{X})} \min_{j \in I_L} L_j(t_{\text{stop}})} \leqslant 1 + \lambda, \tag{74}$$

where $t_{\text{stop}}$ represents the number of rounds after which $\lambda$-SQE halts its execution, we can apply the bounds of Equations (66) and (68) to derive the following stricter condition:

$$\frac{V_* + 2\beta(t_{\text{stop}})}{V_* - 2\beta(t_{\text{stop}})} \leqslant 1 + \lambda, \tag{75}$$

where $\beta(\cdot)$ represents the Chernoff-Hoeffding confidence bound. By rearranging terms, we obtain:

$$\beta(t_{\text{stop}}) \leqslant \frac{\lambda V_*}{2(2 + \lambda)}. \tag{76}$$

By applying the definition of $\beta(\cdot)$ and solving for $t_{\text{stop}}$, we derive:

$$t_{\text{stop}} \geqslant \frac{4(2 + \lambda)^2 \ln\left(\frac{2SAt_{\text{stop}}}{\delta}\right)}{\lambda^2 V_*^2}. \tag{77}$$

Following Lemma 12 of (Jonsson et al., 2020), we can then derive:

$$t_{\text{stop}} \geqslant \frac{8(2 + \lambda)^2}{\lambda^2 V_*^2} \ln\left(\frac{8(2 + \lambda)^2 SA}{\lambda^2 V_*^2 \delta}\right). \tag{78}$$

In order to conclude the proof, it suffices to sum over all queries in $\mathcal{Q}^{(h)}$, accounting for the fact that each query is sampled a number of times that is the minimum between the one in Equation (72), i.e., the query is suboptimal and thus discarded, and the one in Equation (78), i.e., the query is not discarded.

## B. Related Works

In this appendix, we summarize the relevant literature regarding the learning objective and the offline/online approach.

**Learning Objective.** To a first approximation, preference-based approaches can be categorized into two classes, based on their objective: $(i)$ policy optimization and $(ii)$ compatible reward identification.

*Policy optimization* refers to learning a behavior that best aligns with the values of a human labeler. This goal is customarily formalized as maximizing the compliance of the policy with respect to the observed preferences, i.e., maximizing the

probability of generating preferred trajectories. In the literature, this objective is further characterized in two groups based on the adopted approach, namely *reinforcement learning from human feedback* (RLHF, Christiano et al., 2017) and *direct preference optimization* (DPO, An et al., 2023; Azar et al., 2024; Rafailov et al., 2023; Son et al., 2025). The former explicitly aims to recover a *reward model* consistent with the observed preferences, which is then used in pair with a standard RL approach to learn a maximally compliant policy (Zucker et al., 2010; Akrour et al., 2012). The latter bypasses the reward modeling step and instead focuses on directly learning a *control policy*, often constraining the search to policies that can be induced by some underlying reward function and, thus, remaining compliant with the assumption that the human behavior is guided by a reward (Rosset et al., 2024; Nika et al., 2024). Both groups of policy optimization approaches require recovering a reward model (explicit or implicit); however, such a reward model is required to be accurate only in the portion of the state-action space that the optimal policy visits. Intuitively, it suffices that the reward model induces preferences aligned with those of the human, which limits the evaluation of the model only to the trajectories that the human has observed. Such a characteristic, while reasonable for the tasks the cited works address, limits the potential of the learned policy. Coming from the IRL literature (see, e.g., Arora & Doshi, 2021, for a survey on the matter), we have that a learned reward function should enable transferability to a new agent and robustness to slight modifications to the environment. Indeed, a reward model is inherently more transferable than a policy (Russell, 1998). Based upon this reasoning, we formalize and address the *compatible reward identification* problem with respect to RL from preference feedback. Compatible reward identification aims to recover a reward function that is compliant with human preferences *and* accurate across the entire state-action space. One important observation, necessary to keep in mind when comparing results, is that compatible reward identification approaches are prone to having higher query complexity than policy optimization ones. This is due to the higher requirement for accuracy, which in turn provides several guarantees that policy optimization approaches cannot provide.

**Online vs. Offline Approaches.** Works in the RL from preferences literature can further be categorized based on whether they are in the *offline* or *online setting*. In the offline setting, the learning agent must devise a query set a priori of any interaction with the human labeler (see, e.g., Bai et al., 2022; Ouyang et al., 2022). In the online setting, on the other hand, the learning agent can exploit the information it has gathered from previous queries and feedback to make an informed choice on the subsequent query to make (see, e.g., Xie et al., 2025; Razin et al., 2025; Cen et al., 2025; Wu et al., 2025; Liu et al., 2024; Huang et al., 2025; Deng et al., 2025; Feng et al., 2025; Chen et al., 2025; Belakaria et al., 2025; Bai et al., 2025). Since human feedback is typically expensive in real-world applications, in several senses, we aim to *minimize the human query complexity*, i.e., the number of times we ask a human to compare two trajectories. This problem is known as *preference-based exploration* (Shukla & Basu, 2024) and is an instance of the *preference elicitation* problem (Chen & Pu, 2004). Existing works have studied this problem even from a theoretical perspective, primarily in the context of *multi-armed bandits* (MABs, Lattimore & Szepesvári, 2020), and have provided finite-sample/time guarantees on the quality of the recovered reward function or control policy (Wu et al., 2025; Cen et al., 2025; Bai et al., 2025; Feng et al., 2025; Chen et al., 2025). However, when transitioning to the proper RL setting, i.e., sequential decision-making over multiple stages, the problem becomes more challenging due to the *curse of horizon*, and the theoretical understanding remains limited, and the available algorithms work under the BT model only (Wang et al., 2023; Xie et al., 2025).

## C. Reward Identification Algorithm

In this appendix, we propose an algorithm that recovers an $\varepsilon$-optimal reward function $\widehat{r}$ by employing the $\lambda$-`SQE` procedure, and we then provide a numerical validation on synthetic MDPs of different sizes.

### C.1. Algorithm

We now introduce `SuccessiveEliminationRewardEstimation` (`SERE`, Algorithm 3). This algorithm is a *Explore-then-Commit*-like algorithm, as it defines an initial *exploration* phase, in which it runs the $\lambda$-`SQE` procedure to identify a $\lambda$-optimal query chain, and a *commit* phase, in which it samples the queries in the query chain until it can recover an $\varepsilon$-compatible reward w.p. at least $1 - 2\delta$, where $\delta \in (0, 1/2)$.

**Theorem C.1** (Overall Query Complexity). `SERE` *is an* $(\varepsilon, \delta)$-*PAC algorithm with a query complexity:*

$$\widetilde{O} \left( \frac{S^4 A^4 H(2 + \lambda)^2 \ln(1/\delta)}{\varepsilon^2 (2 - 3\lambda/2)^2 V_*^2} + \frac{S^2 A^2 H(2 + \lambda)^2 \ln(1/\delta)}{\lambda^2 V_*^2} \right). \tag{79}$$

---

**Algorithm 3:** `SERE`.

   **Input** :Query set $\mathcal{Q}$, parameters $\varepsilon > 0, \delta > 0, \lambda \in [0, 3/2)$
**1**   $I, \widehat{\boldsymbol{p}}, t \leftarrow \lambda\text{-}\texttt{SQE}(\mathcal{Q}, \lambda, \delta)$
**2**   **for** $i \in I$ **do**
**3**      Sample query $i$ for $\widetilde{O}\left(\frac{(S^3 A^3 \ln(\delta^{-1}))}{\varepsilon^2 (\widehat{V}_I - \beta(t)^2)^2}\right)$ times and compute $\widehat{p}_i$
**4**   **end**

**5**   Compute estimated reward function:      $\widehat{\boldsymbol{r}} = \begin{bmatrix} I \\ \boldsymbol{e}_{SA}^\top \end{bmatrix}^{-1} \begin{bmatrix} \sigma^{-1}(\widehat{\boldsymbol{p}}) \\ 0 \end{bmatrix}$

   **Return :** $\widehat{\boldsymbol{r}}$

---

*Moreover, by selecting* $\lambda = O(\varepsilon/(SA))$, `SERE` *has an overall query complexity bounded by:*

$$\widetilde{O}\left(\frac{S^4 A^4 H \ln(1/\delta)}{\varepsilon^2 V_*^2}\right). \tag{80}$$

One important observation is that, with a convenient selection of $\lambda$, `SERE` is a $(\varepsilon, \delta)$-PAC algorithm with an overall query complexity that is of the same order, up to logarithmic terms, as an algorithm that knows the optimal basis and its index, as shown when comparing Equation (80) with Equation (6).

*Proof.* Recalling the behavior of the $\lambda$-`SQE` algorithm, we divide the query complexity into two stages: *(i)* the *exploration* stage, in which the algorithm identifies a $\lambda$-optimal basis, and *(ii)* the *commit* phase in which the algorithm samples each query in the selected basis a number of times that is sufficient to ensure that Equation (14) holds w.p. at least $1 - \delta$. Thus, we can define the overall query complexity as:

$$K = K_{\text{commit}} + K_{\text{expl}}, \tag{81}$$

where $K_{\text{commit}}$ and $K_{\text{expl}}$ represent the overall query complexity of the commit and exploration phases, respectively.

We bound the overall query complexity of the exploration phase by trivially assuming that each query in $|\mathcal{Q}^{(h)}|$ is not discarded. By combining the bound of Equation (78) with the cardinality of $\mathcal{Q}^{(h)}$, we obtain:

$$K_{\text{expl}} \geqslant \frac{4S^2 A^2 (2 + \lambda)^2}{\lambda^2 V_*^2} \ln\left(\frac{8(2 + \lambda)^2 SA}{\lambda^2 V_*^2 \delta}\right). \tag{82}$$

In order to bound the overall query complexity of the commit phase, we assume to commit to the *worst* chain that can be devised at time $t_{\text{stop}}$, i.e., the one that contains the remaining element, which we will denote as $w$, with the smallest variance, which we will denote as $V_w$. Since query $w$ is part of the basis the algorithm commits to, it has not been discarded during the exploration phase, and thus its discard round is greater than the round at which $\lambda$-`SQE` halts. Thus, we can combine Equations (72) and (78), deriving that:

$$\frac{32}{\Delta_q^2} \ln\left(\frac{32SA}{\Delta_q^2 \delta}\right) \geqslant \frac{8(2 + \lambda)^2}{\lambda^2 V_*^2} \ln\left(\frac{8(2 + \lambda)^2 SA}{\lambda^2 V_*^2 \delta}\right). \tag{83}$$

By defining:

$$f(x) := x \ln\left(\frac{SAx}{\delta}\right), \tag{84}$$

we observe that Equation (83) can be rewritten as:

$$f\left(\frac{32}{\Delta_w^2}\right) \geqslant f\left(\frac{8(2 + \lambda)^2}{\lambda^2 V_*^2}\right). \tag{85}$$

Since $f(\cdot)$ is monotone non-decreasing, then Equation (85) reduces to studying when the argument of the LHS is greater than or equal to the one of the RHS. By applying Definition 5.2, we obtain:

$$\frac{32}{(V_* - V_w)^2} \geqslant \frac{8(2 + \lambda)^2}{\lambda^2 V_*^2}. \tag{86}$$

Isolating for $V_w$, we obtain:

$$V_w \geqslant \frac{2-\lambda}{2+\lambda} V_*. \tag{87}$$

However, $V_w$ is not known to the learner. Thus, we can employ the lower confidence bound to the value of $V_w$ at the end of the exploration phase to define the query complexity of the commit phase, i.e.:

$$\widehat{V}_w(t_{\text{stop}}) - \beta(t_{\text{stop}}). \tag{88}$$

By combining Equation (87), the value of $\beta(\cdot)$ at $t_{\text{stop}}$, and with some algebraic calculations, we derive that:

$$\widehat{V}_w(t_{\text{stop}}) - \beta(t_{\text{stop}}) \geqslant \frac{(2 - 3\lambda/2)V_*}{2(2+\lambda)}. \tag{89}$$

By combining this result with the one of Lemma 5.2, i.e., substituting Equation (89) to $V(\mathcal{Q})$, and considering $R_{\max} \leqslant SA(SA+1)/\pi$, we get:

$$K_{\text{commit}} \geqslant \frac{32eS^4A^4(2+\lambda)^2}{\pi^2\varepsilon^2(2-3\lambda/2)^2V_*^2} \ln\left(\frac{64eS^3A^3(2+\lambda)^2}{\pi^2\varepsilon^2(2-3\lambda/2)^2V_*^2\delta}\right). \tag{90}$$

By combining Equations (82) and (90), and summing over all stages, we obtain the overall sample complexity of the $\lambda$-SQE algorithm. In general, the value of $V_*$ may vary between stages; however, we consider it constant for ease of presentation.

Finally, to find the value of $\lambda$ that minimizes the query complexity, we set $K_{\text{commit}}$ and $K_{\text{expl}}$ to be equal, i.e.:

$$\frac{32eS^4A^4(2+\lambda)^2}{\pi^2\varepsilon^2(2-3\lambda/2)^2V_*^2} \ln\left(\frac{64eS^3A^3(2+\lambda)^2}{\pi^2\varepsilon^2(2-3\lambda/2)^2V_*^2\delta}\right) = \frac{4S^2A^2(2+\lambda)^2}{\lambda^2V_*^2} \ln\left(\frac{8(2+\lambda)^2SA}{\lambda^2V_*^2\delta}\right).$$

For $\varepsilon$ small enough, by selecting $\lambda = O(\varepsilon/SA)$, we trivially observe that the two terms have the same order, up to logarithmic terms. $\qquad\square$

## C.2. Numerical Validation

We now provide a numerical validation of the performance of the SERE, evaluating the impact that the $\varepsilon$ and $\lambda$ hyperparameters have on the performance.

**Setting.** To evaluate the performance of our solution, we consider randomly generated MDPs of varying sizes. For simplicity, we focus on the case where the number of states and actions are equal, i.e., $S = A$. This choice is motivated by the fact that, as discussed in Section 5, the statistical challenge depends on the product $SA$. Consequently, increasing either $S$ or $A$ has a similar impact on performance. We consider $S$ and $A$ both in $\{3, 5, 7\}$. For the horizon, thanks to equivalence classes, we perform estimates for each stage independently. As such, we fix $H = 5$ as an illustrative horizon. Transition probabilities are generated for each environment using (normalized) random uniform probabilities. Rewards are generated following a random uniform distribution in the range $[0, R_{\max}]$, with $R_{\max} \in \{1, 3, 5\}$.

First, we study the impact of parameter $\varepsilon$ on the performance of the policy that can be computed through finite-horizon value iteration using the estimated reward $\widehat{r}$, denoted as $\widehat{\pi}_\varepsilon$. The performance is evaluated in terms of the true reward function $r^*$, comparing the value function of $\widehat{\pi}_\varepsilon$ w.r.t. $r^*$ and the value function of the optimal policy $\pi^*$.

Then, we conduct a sensitivity analysis on parameter $\lambda$, focusing on the variation of ($i$) the index of the basis recovered after the $\lambda$-SQE procedure and of ($ii$) the value of $t_{\text{stop}}$, i.e., the number of rounds after which $\lambda$-SQE reaches the stopping condition.

For each experiment, the results shown are averaged over 5 randomly generated MDPs.

**Impact of Parameter $\varepsilon$.** We report the results of the impact of parameter $\varepsilon$ on the performance of the policy $\widehat{\pi}_\varepsilon$ computed w.r.t. the recovered reward $\widehat{r}$ in Tables 1 and 2. The tables are structured as follows: in the leftmost column, we report the values of $\varepsilon$, and in columns two through four we report the value function of $\widehat{\pi}_\varepsilon$ w.r.t. the true reward function $r^*$, for varying values of $R_{\max}$. In the final row, we report the value function of the optimal policy $V(\pi^*, r^*)$ for comparison. The dimensionality of the MDP is reported in the caption of each table.

| $\varepsilon$ | $V(\widehat{\pi}_\varepsilon; r^*)$ | | |
|---|---|---|---|
| | $R_{\max} = 1$ | $R_{\max} = 3$ | $R_{\max} = 5$ |
| 0.01 | $3.576 \pm 0.201$ | $10.727 \pm 0.602$ | $17.878 \pm 1.003$ |
| 0.5 | $3.576 \pm 0.201$ | $10.727 \pm 0.602$ | $17.878 \pm 1.003$ |
| 1.0 | $3.576 \pm 0.201$ | $10.727 \pm 0.602$ | $17.878 \pm 1.003$ |
| 2.5 | $3.575 \pm 0.202$ | $10.727 \pm 0.602$ | $17.878 \pm 1.003$ |
| 5.0 | $3.573 \pm 0.204$ | $10.726 \pm 0.600$ | $17.857 \pm 1.005$ |
| 7.5 | $3.572 \pm 0.203$ | $10.723 \pm 0.601$ | $17.847 \pm 0.999$ |
| $V(\pi^*, r^*)$ | $3.576 \pm 0.000$ | $10.727 \pm 1.000$ | $17.878 \pm 1.000$ |

*Table 1.* Sensitivity to $\varepsilon$ for $S = A = 3$ (5 runs, mean$\pm$std).

| $\varepsilon$ | $V(\widehat{\pi}_\varepsilon; r^*)$ | | |
|---|---|---|---|
| | $R_{\max} = 1$ | $R_{\max} = 3$ | $R_{\max} = 5$ |
| 0.01 | $3.871 \pm 0.098$ | $11.612 \pm 0.294$ | $19.353 \pm 0.489$ |
| 0.5 | $3.871 \pm 0.098$ | $11.612 \pm 0.294$ | $19.353 \pm 0.489$ |
| 1.0 | $3.871 \pm 0.098$ | $11.612 \pm 0.294$ | $19.353 \pm 0.489$ |
| 2.5 | $3.871 \pm 0.098$ | $11.612 \pm 0.294$ | $19.353 \pm 0.489$ |
| 5.0 | $3.871 \pm 0.098$ | $11.612 \pm 0.294$ | $19.353 \pm 0.489$ |
| 7.5 | $3.871 \pm 0.098$ | $11.612 \pm 0.294$ | $19.353 \pm 0.489$ |
| $V(\pi^*, r^*)$ | $3.871 \pm 0.000$ | $11.612 \pm 0.000$ | $19.353 \pm 0.000$ |

*Table 2.* Sensitivity to $\varepsilon$ for $S = A = 5$ (5 runs, mean$\pm$std).

| $\lambda$ | $V(I)$ | | | | | $t_{\text{stop}}$ |
|---|---|---|---|---|---|---|
| | $h = 1$ | $h = 2$ | $h = 3$ | $h = 4$ | $h = 5$ | |
| 0.1 | $0.243 \pm 0.004$ | $0.242 \pm 0.005$ | $0.247 \pm 0.001$ | $0.246 \pm 0.002$ | $0.246 \pm 0.002$ | $4300 \pm 1699$ |
| 0.3 | $0.240 \pm 0.004$ | $0.241 \pm 0.006$ | $0.247 \pm 0.001$ | $0.245 \pm 0.002$ | $0.244 \pm 0.001$ | $960 \pm 198$ |
| 0.5 | $0.242 \pm 0.004$ | $0.241 \pm 0.004$ | $0.244 \pm 0.002$ | $0.244 \pm 0.002$ | $0.243 \pm 0.002$ | $590 \pm 122$ |
| 1.0 | $0.241 \pm 0.004$ | $0.239 \pm 0.004$ | $0.243 \pm 0.002$ | $0.242 \pm 0.006$ | $0.244 \pm 0.001$ | $342 \pm 59$ |
| $V_*$ | $0.243 \pm 0.004$ | $0.243 \pm 0.005$ | $0.247 \pm 0.001$ | $0.246 \pm 0.002$ | $0.246 \pm 0.002$ | |

*Table 3.* Sensitivity to hyperparameter $\lambda$ for $S = A = 3$ and $R_{\max} = 1$.

As expected, due to the required number of samples defined in Line 3 of Algorithm 3, SERE succeeds in recovering a reward that induces a policy with $\varepsilon$-optimal performance. Moreover, due to the number of samples deriving from a worst-case analysis, we observe that in practice, for any value of $\varepsilon$, the difference between $\widehat{r}$ and $r^*$, and thus the difference in performance, is substantially smaller than $\varepsilon$.

**Sensitivity to Parameter $\lambda$.** We report the results of the sensitivity analysis to parameter $\lambda$ in Tables 3 through 11. Each table reports an experiment run over 5 random instances of an MDP with parameters for $S$, $A$, and $R_{\max}$ reported in the caption of each table. The tables are structured as follows: consider Table 3. The leftmost column reports the considered values of $\lambda$, in particular, we considered $\lambda \in \{0.1, 0.3, 0.5, 1.0\}$. Columns two through six report, for each stage $h \in [\![H]\!]$ the index of matrix $I$, i.e., the matrix selected by SERE after the $\lambda$-SQE procedure is terminated, for each value of $\lambda$, reporting also the index of the optimal basis, i.e., $V_*$, in the last row. Finally, the rightmost column reports the value of $t_{\text{stop}}$. Each value in the tables is reported as the empirical mean computed by averaging over 5 runs and its $95\%$ confidence interval.

| $\lambda$ | $V(I)$ | | | | | $t_{\text{stop}}$ |
| | $h = 1$ | $h = 2$ | $h = 3$ | $h = 4$ | $h = 5$ | |
|---|---|---|---|---|---|---|
| 0.1 | $0.196 \pm 0.025$ | $0.198 \pm 0.029$ | $0.225 \pm 0.006$ | $0.214 \pm 0.017$ | $0.215 \pm 0.013$ | $38300 \pm 40016$ |
| 0.3 | $0.196 \pm 0.025$ | $0.198 \pm 0.029$ | $0.224 \pm 0.006$ | $0.214 \pm 0.017$ | $0.214 \pm 0.014$ | $4928 \pm 4820$ |
| 0.5 | $0.196 \pm 0.025$ | $0.195 \pm 0.028$ | $0.225 \pm 0.006$ | $0.214 \pm 0.017$ | $0.215 \pm 0.013$ | $2120 \pm 1853$ |
| 1.0 | $0.194 \pm 0.023$ | $0.197 \pm 0.029$ | $0.221 \pm 0.007$ | $0.209 \pm 0.015$ | $0.213 \pm 0.014$ | $900 \pm 725$ |
| $V_*$ | $0.196 \pm 0.025$ | $0.198 \pm 0.029$ | $0.225 \pm 0.006$ | $0.214 \pm 0.017$ | $0.215 \pm 0.013$ | |

*Table 4.* Sensitivity to hyperparameter $\lambda$ for $S = A = 3$ and $R_{\max} = 3$.

| $\lambda$ | $V(I)$ | | | | | $t_{\text{stop}}$ |
| | $h = 1$ | $h = 2$ | $h = 3$ | $h = 4$ | $h = 5$ | |
|---|---|---|---|---|---|---|
| 0.1 | $0.138 \pm 0.040$ | $0.144 \pm 0.040$ | $0.188 \pm 0.013$ | $0.170 \pm 0.033$ | $0.170 \pm 0.025$ | $191250 \pm 234407$ |
| 0.3 | $0.138 \pm 0.040$ | $0.144 \pm 0.040$ | $0.188 \pm 0.013$ | $0.170 \pm 0.033$ | $0.170 \pm 0.025$ | $31088 \pm 53745$ |
| 0.5 | $0.138 \pm 0.040$ | $0.144 \pm 0.040$ | $0.188 \pm 0.013$ | $0.170 \pm 0.033$ | $0.169 \pm 0.025$ | $12870 \pm 21980$ |
| 1.0 | $0.138 \pm 0.040$ | $0.142 \pm 0.039$ | $0.188 \pm 0.013$ | $0.169 \pm 0.033$ | $0.168 \pm 0.024$ | $4560 \pm 7653$ |
| $V_*$ | $0.138 \pm 0.040$ | $0.144 \pm 0.040$ | $0.188 \pm 0.013$ | $0.170 \pm 0.033$ | $0.170 \pm 0.025$ | |

*Table 5.* Sensitivity to hyperparameter $\lambda$ for $S = A = 3$ and $R_{\max} = 5$.

| $\lambda$ | $V(I)$ | | | | | $t_{\text{stop}}$ |
| | $h = 1$ | $h = 2$ | $h = 3$ | $h = 4$ | $h = 5$ | |
|---|---|---|---|---|---|---|
| 0.1 | $0.248 \pm 0.0$ | $0.248 \pm 0.001$ | $0.248 \pm 0.001$ | $0.248 \pm 0.0$ | $0.248 \pm 0.0$ | $2600 \pm 297$ |
| 0.3 | $0.247 \pm 0.001$ | $0.246 \pm 0.001$ | $0.247 \pm 0.001$ | $0.248 \pm 0.0$ | $0.247 \pm 0.001$ | $800 \pm 0$ |
| 0.5 | $0.246 \pm 0.001$ | $0.245 \pm 0.001$ | $0.247 \pm 0.001$ | $0.246 \pm 0.001$ | $0.246 \pm 0.001$ | $500 \pm 0$ |
| 1.0 | $0.245 \pm 0.002$ | $0.244 \pm 0.002$ | $0.245 \pm 0.003$ | $0.244 \pm 0.001$ | $0.246 \pm 0.002$ | $300 \pm 0$ |
| $V_*$ | $0.248 \pm 0.0$ | $0.248 \pm 0.001$ | $0.248 \pm 0.001$ | $0.249 \pm 0.0$ | $0.249 \pm 0.0$ | |

*Table 6.* Sensitivity to hyperparameter $\lambda$ for $S = A = 5$ and $R_{\max} = 1$.

| $\lambda$ | $V(I)$ | | | | | $t_{\text{stop}}$ |
| | $h = 1$ | $h = 2$ | $h = 3$ | $h = 4$ | $h = 5$ | |
|---|---|---|---|---|---|---|
| 0.1 | $0.236 \pm 0.003$ | $0.232 \pm 0.008$ | $0.236 \pm 0.004$ | $0.241 \pm 0.002$ | $0.24 \pm 0.003$ | $8600 \pm 3469$ |
| 0.3 | $0.235 \pm 0.003$ | $0.23 \pm 0.009$ | $0.235 \pm 0.004$ | $0.24 \pm 0.002$ | $0.239 \pm 0.003$ | $1456 \pm 382$ |
| 0.5 | $0.235 \pm 0.003$ | $0.231 \pm 0.008$ | $0.235 \pm 0.004$ | $0.239 \pm 0.003$ | $0.239 \pm 0.004$ | $810 \pm 112$ |
| 1.0 | $0.233 \pm 0.006$ | $0.227 \pm 0.008$ | $0.234 \pm 0.004$ | $0.237 \pm 0.007$ | $0.239 \pm 0.003$ | $444 \pm 45$ |
| $V_*$ | $0.237 \pm 0.003$ | $0.233 \pm 0.008$ | $0.236 \pm 0.005$ | $0.241 \pm 0.002$ | $0.24 \pm 0.003$ | |

*Table 7.* Sensitivity to hyperparameter $\lambda$ for $S = A = 5$ and $R_{\max} = 3$.

Some comments are in order. First, we observe that the index of the optimal basis $V_*$ is directly proportional to the dimensionality of the MDP. This is due to the fact that, having generated the true rewards by sampling from a random uniform distribution, a higher number of states and actions translates into a higher number of possible queries, i.e., state-action pairs, and thus, a higher probability of the existence of a basis with a greater minimum variance.

Then, we observe that the index of the optimal basis is inversely proportional to the maximum reward $R_{\max}$. This is due to

| $\lambda$ | $V(I)$ | | | | | $t_{\text{stop}}$ |
| --- | --- | --- | --- | --- | --- | --- |
| | $h = 1$ | $h = 2$ | $h = 3$ | $h = 4$ | $h = 5$ | |
| 0.1 | $0.215 \pm 0.007$ | $0.206 \pm 0.019$ | $0.215 \pm 0.011$ | $0.226 \pm 0.005$ | $0.223 \pm 0.007$ | $25550 \pm 15064$ |
| 0.3 | $0.214 \pm 0.007$ | $0.205 \pm 0.019$ | $0.214 \pm 0.011$ | $0.226 \pm 0.004$ | $0.223 \pm 0.008$ | $3504 \pm 1813$ |
| 0.5 | $0.215 \pm 0.007$ | $0.205 \pm 0.019$ | $0.213 \pm 0.011$ | $0.225 \pm 0.005$ | $0.223 \pm 0.008$ | $1580 \pm 716$ |
| 1.0 | $0.212 \pm 0.008$ | $0.205 \pm 0.021$ | $0.213 \pm 0.01$ | $0.225 \pm 0.004$ | $0.223 \pm 0.007$ | $654 \pm 240$ |
| $V_*$ | $0.215 \pm 0.007$ | $0.207 \pm 0.019$ | $0.215 \pm 0.011$ | $0.226 \pm 0.005$ | $0.223 \pm 0.008$ | |

*Table 8.* Sensitivity to hyperparameter $\lambda$ for $S = A = 5$ and $R_{\max} = 5$.

| $\lambda$ | $V(I)$ | | | | | $t_{\text{stop}}$ |
| --- | --- | --- | --- | --- | --- | --- |
| | $h = 1$ | $h = 2$ | $h = 3$ | $h = 4$ | $h = 5$ | |
| 0.1 | $0.249 \pm 0.0$ | $0.248 \pm 0.001$ | $0.249 \pm 0.0$ | $0.248 \pm 0.001$ | $0.249 \pm 0.0$ | $2500 \pm 0$ |
| 0.3 | $0.248 \pm 0.001$ | $0.248 \pm 0.0$ | $0.248 \pm 0.0$ | $0.248 \pm 0.001$ | $0.248 \pm 0.0$ | $800 \pm 0$ |
| 0.5 | $0.247 \pm 0.001$ | $0.247 \pm 0.001$ | $0.247 \pm 0.0$ | $0.247 \pm 0.001$ | $0.247 \pm 0.001$ | $500 \pm 0$ |
| 1.0 | $0.245 \pm 0.001$ | $0.244 \pm 0.002$ | $0.245 \pm 0.001$ | $0.246 \pm 0.001$ | $0.245 \pm 0.001$ | $300 \pm 0$ |
| $V_*$ | $0.249 \pm 0.001$ | $0.249 \pm 0.0$ | $0.25 \pm 0.0$ | $0.249 \pm 0.0$ | $0.249 \pm 0.0$ | |

*Table 9.* Sensitivity to hyperparameter $\lambda$ for $S = A = 7$ and $R_{\max} = 1$.

| $\lambda$ | $V(I)$ | | | | | $t_{\text{stop}}$ |
| --- | --- | --- | --- | --- | --- | --- |
| | $h = 1$ | $h = 2$ | $h = 3$ | $h = 4$ | $h = 5$ | |
| 0.1 | $0.243 \pm 0.004$ | $0.244 \pm 0.002$ | $0.246 \pm 0.001$ | $0.245 \pm 0.001$ | $0.244 \pm 0.005$ | $4500 \pm 1725$ |
| 0.3 | $0.242 \pm 0.004$ | $0.243 \pm 0.003$ | $0.245 \pm 0.0$ | $0.243 \pm 0.001$ | $0.244 \pm 0.004$ | $1072 \pm 237$ |
| 0.5 | $0.241 \pm 0.006$ | $0.242 \pm 0.003$ | $0.244 \pm 0.001$ | $0.243 \pm 0.001$ | $0.241 \pm 0.004$ | $620 \pm 125$ |
| 1.0 | $0.241 \pm 0.004$ | $0.238 \pm 0.004$ | $0.238 \pm 0.004$ | $0.241 \pm 0.001$ | $0.241 \pm 0.004$ | $360 \pm 64$ |
| $V_*$ | $0.243 \pm 0.004$ | $0.244 \pm 0.002$ | $0.246 \pm 0.001$ | $0.245 \pm 0.001$ | $0.245 \pm 0.004$ | |

*Table 10.* Sensitivity to hyperparameter $\lambda$ for $S = A = 7$ and $R_{\max} = 3$.

| $\lambda$ | $V(I)$ | | | | | $t_{\text{stop}}$ |
| --- | --- | --- | --- | --- | --- | --- |
| | $h = 1$ | $h = 2$ | $h = 3$ | $h = 4$ | $h = 5$ | |
| 0.1 | $0.232 \pm 0.011$ | $0.234 \pm 0.006$ | $0.238 \pm 0.002$ | $0.236 \pm 0.003$ | $0.236 \pm 0.011$ | $10200 \pm 6172$ |
| 0.3 | $0.231 \pm 0.011$ | $0.233 \pm 0.005$ | $0.238 \pm 0.002$ | $0.236 \pm 0.002$ | $0.235 \pm 0.011$ | $1616 \pm 676$ |
| 0.5 | $0.232 \pm 0.011$ | $0.232 \pm 0.008$ | $0.237 \pm 0.003$ | $0.236 \pm 0.003$ | $0.235 \pm 0.013$ | $880 \pm 271$ |
| 1.0 | $0.231 \pm 0.011$ | $0.231 \pm 0.006$ | $0.233 \pm 0.004$ | $0.233 \pm 0.005$ | $0.233 \pm 0.011$ | $474 \pm 60$ |
| $V_*$ | $0.232 \pm 0.011$ | $0.234 \pm 0.006$ | $0.238 \pm 0.002$ | $0.237 \pm 0.002$ | $0.236 \pm 0.011$ | |

*Table 11.* Sensitivity to hyperparameter $\lambda$ for $S = A = 7$ and $R_{\max} = 5$.

the underlying BT model that defines the preference probability. In fact, increasing the scale of the rewards increases also the scale of the difference between the reward of state-action pairs. This, in turn, translates into preference probabilities closer to the extremes, and thus in a lower variance of the preference generation.

Moreover, we observe that the index of the recovered basis $V(I)$ is inversely proportional to $\lambda$. This behavior is expected, as a higher $\lambda$ corresponds to a larger stopping condition in $\lambda$-SQE, and thus less exploration. As a consequence, more queries

|         | $SA$ | $R_{\max}$ | $\lambda$ |
|---------|:----:|:----------:|:---------:|
| $V_*$       | ↑ | ↓ | — |
| $V(I)$      | — | — | ↓ |
| $t_{\text{stop}}$ | — | ↑ | ↓ |

*Table 12.* Summary of the theoretical and empirical findings.

are feasible to recover $I$, and a looser confidence interval may lead to recovering a suboptimal basis.

Finally, we observe that the number of rounds $t_{\text{stop}}$ after which $\lambda$-SQE reaches the stopping condition is inversely proportional to $\lambda$, for the same reasons as the previous observation, and directly proportional to $R_{\max}$. Following previous observations, we notice that the value $\underline{V}(I_L)$ in Line 22 of Algorithm 1 tends to 0 as $R_{\max}$ increases, thus increasing the number of required rounds, with $\lambda$ being equal.

**Summary of Numerical Validation.** We now summarize the results and observations of the numerical validation.

First, we observe that the SERE algorithm succeeds in recovering a reward that induces a policy whose performance is close to the optimal one. Moreover, the difference between the true and the recovered rewards is much smaller than the value of parameter $\varepsilon$, due to the worst-case analysis we conducted.

Then, we observe that the sample complexity of the $\lambda$-SQE procedure, and the suboptimality of the selected basis, depend on both the dimensionality of the MDP and on the selected value of $\lambda$. We report an aggregated summary of such dependencies in Table 12, in which we show the direct (i.e., ↑) or inverse (i.e., ↓) proportionality of the parameters in the rows w.r.t. the parameters in the columns.

