# OpenReview forum: "Online Compatible Reward Identification from Preference Feedback"
_ICML.cc/2026/Conference — ICML 2026 regular_

### Official Review · Reviewer_u3ap · 2026-02-28

**Soundness:** 3
**Presentation:** 3
**Significance:** 2
**Originality:** 3
**Overall Recommendation:** 4
**Confidence:** 3

**Summary:**

This work focuses on how to efficiently recover the reward function based on preference feedback.

The paper first formulates the Compatible Reward Identification from Trajectory Preferences (CRI-Tf) problem in finite-horizon MDPs (Sec. 2). And describes the queries and learning problem (Sec. 3), studying the minimum number of queries that can recover the set of rewards. For deterministic preferences, the authors prove lower bounds ($HSA(SA -1)/2$ for offline and $\Omega HSA\log(SA)$ for online, Theorems 4.3-4.4) and inapproximability results which demonstrate that using fewer than sufficient data might lead to great error (Theorem 4.2). For stochastic preferences modeled (the Bradley–Terry model), the paper proposes algorithms to identify $\lambda$-optimal query chains using successive elimination and provide PAC guarantees for recovering a compatible reward (Theorems 5.4–5.5) .

Overall, the paper provides a theoretical framework for reward recovery from preferences, including lower bounds and algorithmic procedures.

**Compliance With Llm Reviewing Policy:**

Affirmed.

**Final Justification:**

After reading the paper and the authors’ response, I think the paper has technical interest for identifying the reward model. I am positive about this paper. The work offers a novel and technically interesting idea. The paper is also generally well written, and the theoretical results support the central claims reasonably well. I maintain the score 4.

**Key Questions For Authors:**

- What would be the result if combining this paper with the RLHF work?

- How does the computational complexity of λ-SQE as a function of $S, A, H$ compare to the query complexity?

- In RLHF, when you train the policy, you will also maintain a reward model. Could you elaborate more on the difference in estimating reward between RLHF and this paper.

**Limitations:**

yes

**Strengths And Weaknesses:**

- Strength

The paper is well structured, from problem formulation (Sec. 2–3) to deterministic analysis (Sec. 4), stochastic analysis (Sec. 5), and conclusions (Sec. 6). Key concepts such as compatible reward sets, query matrices, and Hausdorff evaluation are defined clearly. The formulation of compatible reward identification distinguishes this work from policy-focused preference learning. The query index and its connection to the BT model provide a perspective on query selection (Sec. 5). The characterization that queries with index near 1/4 are most informative is a nontrivial insight (Sec. 6).

- Weakness: refer to the Question section.

---

> ### Author Rebuttal · Authors · 2026-03-31
>
> We thank the Reviewer for the time spent reviewing our work. Below, our answers to the Reviewer's questions and concerns.
>
> > What would be the result if combining this paper with the RLHF work?
>
> We would like to clarify that the goal of our paper is not to provide a practical or large-scale RLHF algorithm, but rather to study the inherent challenges of the specific problem of identifying a reward function from preferences. That said, as is customary in this literature, we believe that the tabular setting is, in any case, a necessary step toward understanding how to tackle more complex scenarios in a principled way.
> Nevertheless, we believe that our approach can be integrated into an RLHF pipeline. Its role would be to suggest which trajectories the RLHF algorithm should sample from the environment in order to minimize the cost of querying the human for preferences (what we refer to as query complexity). In terms of results, we expect, at least in practice, faster convergence to a compatible reward function and a reduction in the number of preferences requested from the human. Nevertheless, we believe that such an integration into an RLHF pipeline is outside the scope of the present paper.
>
> > How does the computational complexity of $\lambda$-SQE as a function of $S, A, H$ compare to the query complexity?
>
> Considering the cardinality of the initial query set as $SA(SA-1)$ for every stage, we can see that the main computational complexity of the $\lambda$-SQE algorithm lies in its BestBasis procedure, which is invoked for every query, and iterates over all queries. Thus this induces a computational complexity that, w.r.t. $S, A$, and $H$ only, is $O(S^4 A^4 H)$, as the procedure is repeated until a basis is found and for every stage.
> Crucially, however, the computational complexity of $\lambda$-SQE and its query complexity are orthogonal. Indeed, the query complexity refers to the number of interactions with the human labeler, and it is what we effectively want to minimize; the computational complexity, on the other hand, refers to the agent only, and we only need it to be polynomial in the relevant terms to make the algorithm actionable.
>
> > In RLHF, when you train the policy, you will also maintain a reward model. Could you elaborate more on the difference in estimating reward between RLHF and this paper.
>
> Indeed, in RLHF, the estimation of the reward model and the optimization of the policy are deeply connected, as the policy optimization objective requires that the reward function is accurate in that portion of the state-action space visited by the optimal policy.
> The main difference between RLHF and CRI lies exactly in this point. Our objective is to recover a reward function that is accurate across the entire state-action space. This brings advantages in scenarios such as the sim-to-real, in which the real-world dynamics of the agent and of the environment may differ from those of the simulation, and thus the simulation's optimal policy might be unsafe and suboptimal in the real world. Instead, CRI avoids this problem entirely, as it allows the agent to adapt without requiring any additional interaction with a human. We will add a discussion on this in the paper to further clarify this distinction.

---

> > ### Author Rebuttal · Reviewer_u3ap · 2026-04-02
> >
> > Thanks for the detailed response. The response resolved my concerns and I maintain my original “Accept” score.

---

### Official Review · Reviewer_Gaay · 2026-03-04

**Soundness:** 3
**Presentation:** 3
**Significance:** 2
**Originality:** 3
**Overall Recommendation:** 4
**Confidence:** 2

**Summary:**

This paper studies the problem of learning a reward function from preference feedback in both deterministic and stochastic preference settings. The proposed algorithm learns a reward function that is not only consistent with the observed preference feedback but also generalizes across the entire state–action space. By introducing the concept of a query basis and its associated index, the authors derive a bound on the query complexity required for reward identification.

**Compliance With Llm Reviewing Policy:**

Affirmed.

**Final Justification:**

My concerns have been resolved, and I will keep my score unchanged.

**Key Questions For Authors:**

1. Does the proposed algorithm extend to continuous environments? In particular, do the theoretical guarantees still hold when the state or action spaces are continuous rather than tabular?
2. The empirical validation is conducted in relatively small environments (e.g., the number of states and actions is at most 7). How does the proposed algorithm scale to larger environments with significantly higher-dimensional state and action spaces?
3. If the number of sampled preference queries does not meet the theoretical lower bound, how far could the learned reward function deviate from the compatible reward set?

**Limitations:**

Yes

**Strengths And Weaknesses:**

Strengths:
1. The paper studies the problem of minimizing query sample complexity for reward identification and provides theoretical bounds on the required number of queries.
2. The theoretical formulation appears rigorous and well-grounded, providing formal support for the proposed method.

Weaknesses:
1. The proposed framework focuses on tabular environments, whereas many real-world reinforcement learning problems involve continuous state and action spaces. The applicability of the method in such settings remains unclear.
2. The empirical validation is presented only in the appendix rather than in the main body of the paper.

---

> ### Author Rebuttal · Authors · 2026-03-31
>
> We thank the Reviewer for the time spent reviewing our work. Below, our answers to the Reviewer's questions and concerns.
>
> > The proposed framework focuses on tabular environments, whereas many real-world reinforcement learning problems involve continuous state and action spaces. The applicability of the method in such settings remains unclear.
>
> > Does the proposed algorithm extend to continuous environments? In particular, do the theoretical guarantees still hold when the state or action spaces are continuous rather than tabular?
>
> > The empirical validation is conducted in relatively small environments (e.g., the number of states and actions is at most 7). How does the proposed algorithm scale to larger environments with significantly higher-dimensional state and action spaces?
>
> We would like to clarify that the goal of our paper is not to provide a practical/large-scale algorithm, but rather to study the inherent challenges of the specific problem of identifying a reward function from preferences. That said, as is customary in this literature, we believe that the tabular setting is, in any case, a necessary step toward understanding how to tackle more complex/real-world scenarios in a principled way. We will clarify this in both the abstract and the introduction of the paper.
>
> Our bound scales polynomially with the cardinality of the state and action spaces. Therefore, a direct application to very large MDPs is infeasible, which is a common limitation of algorithms designed for the tabular setting. However, in such cases, following well-established RL approaches, one possible solution is to employ a feature representation of the reward function; for example, defining $r(s,a) = \theta^{\top} \phi(s,a)$, where $\theta \in \mathbb{R}^d$ is the unknown parameter we aim to estimate, and $\phi: \mathcal{S} \times \mathcal{A} \to \mathbb{R}^d$ is a known feature mapping.
> Using such an approach, and similarly to standard RL/IRL settings, we believe it may be possible to eliminate the dependence on the cardinality of the state and action spaces, replacing it with a dependence on the feature dimension $d$, which is typically much smaller in practical problems.
> We will add a discussion of this idea, along with some intuition on how to proceed, in the Future Work section of our paper.
>
>
> > The empirical validation is presented only in the appendix rather than in the main body of the paper.
>
> Indeed, we had to make the choice of deferring the empirical validation to the appendix due to the page constraint. We will use the additional page of the camera-ready to add a summary of the experimental validation in the main paper.
>
> > If the number of sampled preference queries does not meet the theoretical lower bound, how far could the learned reward function deviate from the compatible reward set?
>
> In principle, if the query set employed is non-sufficient, Theorem 4.2 holds, showing that the error can be arbitrarily large. The more interesting case, which we believe is the one the Reviewer is referring to, is that of using a sufficient query set but with too few samples per query. Having fixed a query chain, from Lemma 5.2, suppose that we consider a number of samples that is a fraction $\alpha \in (0,1)$ of those prescribed by Lemma 5.2; we then obtain a corresponding guarantee equivalent to an $\epsilon/\sqrt{\alpha}$ accurate compatible reward set (i.e., less accurate). However, there is an additional consideration to make. Suppose instead that we commit to a query basis too soon. In this case, the estimation error also depends on the index of the selected basis and can therefore be amplified.

---

> > ### Author Rebuttal · Reviewer_Gaay · 2026-04-02
> >
> > Thank you to the authors for the rebuttal. I have no further questions.

---

### Official Review · Reviewer_bU86 · 2026-03-10

**Soundness:** 3
**Presentation:** 3
**Significance:** 3
**Originality:** 3
**Overall Recommendation:** 4
**Confidence:** 4

**Summary:**

This paper tries to address an important problem in preference-based RL in how to identify compatible reward functions from human preference feedback. The authors derive the minimum number of interactions required for deterministic preferences, and also propose a (ϵ,δ)-PAC algorithm with polynomial query complexity for the stochastic preference case under Bradley-Terry model. The paper also tries to differentiate this task from standard policy optimization, emphasizing that accuracy across the entire state-action space is important for transferability and safety. The problem formulation is interesting, and the theoretical contribution is meaningful.

**Compliance With Llm Reviewing Policy:**

Affirmed.

**Ethical Review Concerns:**

None.

**Key Questions For Authors:**

1. How feasible is the assumption that arbitrary trajectories can be constructed, especially for long horizon problems?
2. Does the polynomial query complexity remain practical for environments with very large state-action spaces, even within the tabular setting?

**Limitations:**

The analysis only considers BT or deterministic preference models, and robustness to model misspecification is not tested. Also polynomial complexity can still be very large in practice, which needs more discussion on when this is actually tractable. Experiments are synthetic only, there is no evidence on real downstream policy performance.

**Strengths And Weaknesses:**

- soundness
Strength: The paper provides definitions for sufficient, minimal, and minimum query sets, and establishes theoretical bounds for query complexity in both deterministic and stochastic settings. The theoretical framework is generally sound and the derivations seem correct.
Weakness: However, one assumption that is confusing is that trajectory generation can be done "at will." This is quite strong assumption and reduces the applicability in realistic RL settings. In practice we usually cannot freely construct arbitrary trajectories, especially in complex or safety-critical environment.
- presentation
The paper is quite dense and sometimes difficult to follow, mainly because of the heavy notation load. Also, some key details, like specific algorithm implementations and numerical results, are placed in appendix, which make the main text feel incomplete and not self-contained.
- significance
Strength: This paper provides foundational bounds for an underexplored objective in preference-based RL. The distinction between reward identification and policy optimization is a useful conceptual contribution.
Weakness: The lack of experiments on real RL tasks is a weakness. Without showing how these theoretical bounds translate to actual performance in downstream tasks, it is hard to judge the practical impact of this work. The significance feel more potential than demonstrated at this point.
- originality
The originality is a strong aspect of this paper. Introducing new theoretical complexity results specifically for preference-based reward identification is a fresh angle, and the connection between structural properties of query sets, like bases and chains, with statistical efficiency is interesting and novel.

---

> ### Author Rebuttal · Authors · 2026-03-31
>
> We thank the Reviewer for the time spent reviewing our work. Below, our answers to the Reviewer's questions and concerns.
>
> > How feasible is the assumption that arbitrary trajectories can be constructed, especially for long horizon problems?
>
> In the spirit of our work, we consider the most expensive cost to be that of asking a human to provide a comparison between two trajectories, while treating the cost of generating two trajectories as (almost) negligible. Under this perspective, the assumption serves to decouple the statistical complexity of learning (i.e., human query complexity) from the complexity of exploring the environment (Azar et al., 2013). These two problems are orthogonal, and our focus is on addressing the former.
> As an intuitive, heuristic illustration of how both aspects could be handled together, suppose our algorithm requests a trajectory $\tau$. One could then compute or learn a policy that maximizes the probability of generating $\tau$, and interact with the environment until it is produced (or, even more naively, generate trajectories with a uniformly at random policy or a baseline safe policy). Clearly, this may incur in a large sample complexity in terms of environment interactions. Crucially, however, this cost is decoupled from the human query complexity, which is the focus of our work, and thus remains unaffected.
>
> > Also, some key details, like specific algorithm implementations and numerical results, are placed in appendix, which make the main text feel incomplete and not self-contained.
>
> Some relevant details are deferred to the appendix because of the page limits. We will use the additional page in the camera-ready to bring a summary of the experimental evaluation in the main paper.
>
> > The lack of experiments on real RL tasks is a weakness. Without showing how these theoretical bounds translate to actual performance in downstream tasks, it is hard to judge the practical impact of this work. The significance feel more potential than demonstrated at this point.
>
> We note that we provide an experiment on an RL task in Appendix C.2. Since the algorithm we propose is designed for tabular MDPs, the experiment is conducted in this setting. In particular, we show that our algorithm is able to recover a reward function that can be used to learn a policy whose performance is close to optimal. In this sense, the experiment demonstrates the effectiveness of the approach for the setting considered in the paper. As also mentioned in the answer to Reviewer kAc6, the goal of our paper is not to provide a practical/large-scale RLHF algorithm, but rather to study the inherent challenges of the specific problem of identifying a reward function from preferences.
>
> > Does the polynomial query complexity remain practical for environments with very large state-action spaces, even within the tabular setting?
>
> Our bound scales polynomially with the cardinality of the state and action spaces. Therefore, a direct application to very large MDPs is intractable, which is a common limitation of algorithms designed for the tabular setting. However, in such cases, following well-established RL approaches, one possible solution is to employ a feature representation of the reward function; for example, defining $r(s,a) = \theta^{\top} \phi(s,a)$, where $\theta \in \mathbb{R}^d$ is the unknown parameter we aim to estimate, and $\phi: \mathcal{S} \times \mathcal{A} \to \mathbb{R}^d$ is a known feature mapping.
> Using such an approach, and similarly to standard RL/IRL settings, we believe it may be possible to eliminate the dependence on the cardinality of the state and action spaces, replacing it with a dependence on the feature dimension $d$, which is typically much smaller in practical problems.
> We will add a discussion of this idea, along with some intuition on how to proceed, in the Future Work section of our paper.
>
> **References**
>
> Gheshlaghi Azar, M., Munos, R. and Kappen, H.J., 2013. Minimax PAC bounds on the sample complexity of reinforcement learning with a generative model. Machine learning, 91(3), pp.325-349.

---

> > ### Author Rebuttal · Reviewer_bU86 · 2026-04-03
> >
> > The rebuttal addressed my comments. I will keep my positive rating as is.

---

### Official Review · Reviewer_kAc6 · 2026-03-12

**Soundness:** 2
**Presentation:** 3
**Significance:** 2
**Originality:** 3
**Overall Recommendation:** 4
**Confidence:** 3

**Summary:**

This paper is mainly theory-oriented and formalizes the problem of compatible reward identification from preference feedback in a tabular RL setting. The authors derive query-complexity results in both deterministic and stochastic (Bradley–Terry) settings, and provide a PAC guarantee for recovering an $\varepsilon$-compatible reward with high probability.
Personally, Theorem 4.2 (Inapproximability), which shows that if the query set is not sufficient, then even approximate recovery can fail in the worst case, is particularly interesting and could have meaningful impact on the community.

**Compliance With Llm Reviewing Policy:**

Affirmed.

**Final Justification:**

My concerns have been mostly addressed, so I maintain a positive view of the paper.

**Key Questions For Authors:**

1. Could the authors provide a more concrete real-world motivation for this setting?

2. Why is the compatible reward important in RLHF? If the claimed benefit is improved transferability or robustness, it would be helpful to support this more directly, even with a small tabular experiment.

**Limitations:**

yes

**Strengths And Weaknesses:**

### Strengths

Overall, this paper is a solid theory paper. The problem formulation is clear, and the results seem correct and meaningful.


### Weaknesses

1. A main weakness of this paper is that the problem setting seems somewhat stylized, especially given the RLHF motivation. In particular, the analysis is carried out in a tabular setting, which makes the practical relevance of the proposed algorithm somewhat unclear. It would have been better to at least propose a practical algorithm that could be applied to LLMs.

2. The numerical validation is fairly limited. The experiments are conducted only on very small tabular MDPs (S=A=5) and do not include comparisons against meaningful baselines, such as random query selection. Moreover, the experimental results should be presented in the main paper (not only in the Appendix).


3. The paper seems closer to a tabular inverse bandit or preference-based IRL formulation than to modern large-scale RLHF settings. The authors should make this distinction clearer.

---

> ### Author Rebuttal · Authors · 2026-03-31
>
> We thank the Reviewer for the time spent reviewing our work. Below, our answers to the Reviewer's questions and concerns.
>
> > A main weakness of this paper is that the problem setting seems somewhat stylized, especially given the RLHF motivation. In particular, the analysis is carried out in a tabular setting, which makes the practical relevance of the proposed algorithm somewhat unclear. It would have been better to at least propose a practical algorithm that could be applied to LLMs.
>
> > The paper seems closer to a tabular inverse bandit or preference-based IRL formulation than to modern large-scale RLHF settings. The authors should make this distinction clearer.
>
> We would like to clarify that the goal of our paper is not to provide a practical/large-scale RLHF algorithm, but rather to study the inherent challenges of the specific problem of identifying a reward function from preferences. This is indeed connected to IRL from preferences. Our objective is to identify the complete compatible reward set, as is done for the traditional (no preferences) IRL setting in (Metelli et al., 2023), rather than a single reward function. This represents a fundamental step in several RLHF pipelines, and in this sense, the problem we study is motivated by RLHF. We will clarify this in both the abstract and the introduction of the paper.
> That said, as is customary in this literature, we believe that the tabular setting is, in any case, a necessary step toward understanding how to tackle more complex scenarios in a principled way.
>
> > Could the authors provide a more concrete real-world motivation for this setting?
>
> > Why is the compatible reward important in RLHF? If the claimed benefit is improved transferability or robustness, it would be helpful to support this more directly, even with a small tabular experiment.
>
> A motivating example is that of sim-to-real robotics. Indeed, considering a robotic task, human feedback can be employed to optimize a policy in a simulated environment. However, when deploying the agent to the real world, the dynamics of the robot may differ, as well as the observations may be more noisy than in the simulation. If a PbRL approach for policy optimization is only accurate on the simulated distribution of the rewards, when moving to the real world the agent may show degraded behavior. Compatible reward identification, instead, allows reconstructing the entire reward function, thus improving the sim-to-real process and allowing for better performance in the real-world scenario. We will add the example in the final version of the paper.
>
>
> > The numerical validation is fairly limited. The experiments are conducted only on very small tabular MDPs (S=A=5) and do not include comparisons against meaningful baselines, such as random query selection. Moreover, the experimental results should be presented in the main paper (not only in the Appendix).
>
> We thank the Reviewer for raising this point. Indeed, the experimental evaluation of our work is in the appendix due to the space constraints. We will use the additional page in the camera-ready to include a discussion on the experiment in the main paper.
>
> For completeness, as suggested by the Reviewer, we implemented a comparison between our algorithm and a baseline which works on a random basis. We tested two configurations with $S=A=3$ and $S=A=5$, both with $H=5$ and $R_{\max} = 1$. We averaged the results over 5 seeds. We evaluate the improvement in terms of the total samples ratio $T_{\text{random}} / T_{\text{SERE}}$, where $T_{\text{random}}$ and $T_{\text{SERE}}$ are the number of samples to recover the set of $\epsilon$-compatible ($\epsilon=10^{-2}$) rewards with a random basis and the one suggested by our algorithm (including the samples needed to learn such a basis), respectively.
>
> |     $\ $    | | $S=A=3$ | $S=A=5$ |
> | --------- | - | --------- | --------- |
> | $T_{\text{random}} / T_{\text{SERE}}$ | | $4.33 \pm 0.34$ | $3.47 \pm 0.61$ |
>
> The reported result shows that the baseline algorithm, which commits to a random query basis in the beginning, requires a larger number of sample than SERE.
> Indeed, in SERE, the algorithm initially identifies a $\lambda$-optimal basis. The selection of $\lambda$ defines a trade-off between the first and the second phase of the algorithm. Notably, however, as also shown empirically, the improvement provided in the second phase by selecting a good basis in the first phase outweighs the cost of finding it.
> This is due to the sample complexity bound reported in Eq. (4), which shows that, in order to guarantee an error lower than $\epsilon$, we require a number of samples that depends on $1/V(\mathcal{Q})^2$. Our experiment shows that committing to a random basis yields a basis index that is much worse than a $\lambda$-optimal one, thus increasing the sample complexity.
>
> **References**
>
> Metelli, A. M., Lazzati, F., and Restelli, M. Towards theoretical understanding of inverse reinforcement learning. ICML, 2023.

---

> > ### Author Rebuttal · Reviewer_kAc6 · 2026-04-01
> >
> > Thank you for the detailed response. My concerns have been largely addressed, and I will maintain my positive score.

---

### Decision · Program_Chairs · 2026-04-30

**Decision:**

Accept (regular)

**Comment:**

The submitted paper considers compatible reward identification from trajectory preferences and provides query-complexity results in both deterministic and Bradley–Terry (BT) stochastic settings: it introduces the concepts of query bases and their indices, proves lower bounds and an inapproximability result when queries are insufficient, and gives a PAC procedure with polynomial human query complexity. All reviewers recommend acceptance and indicate that their concerns were resolved: while they initially had concerns regarding the "stylized" tabular scope, the strong generate trajectories at will assumption, scalability to large or continuous spaces, and positioning relative to RLHF/IRL, the authors clarified the work's foundational aim, and several other aspects, e.g., concrete motivation (e.g., sim-to-real) and the separation of human query complexity from environment interaction cost. The reviewers considered the rebuttal to resolve their concerns and hence, in line with the reviewers' concensus, I am recommending acceptance of the paper.
For the camera-ready version of the paper, the authors should update their paper in line with the reviewers suggestions and questions and in particular: (i) move and expand the empirical section (including the random-query baseline) to the main text, (ii) clarify assumptions, discuss robustness/misspecification, and (iii) briefly detail computational vs. query complexity.